# Robust recognition and exploratory analysis of crystal structures via Bayesian deep learning

Andreas Leitherer [1✉], Angelo Ziletti[1] & Luca M. Ghiringhelli [1]

Due to their ability to recognize complex patterns, neural networks can drive a paradigm shift in the analysis of materials science data. Here, we introduce ARISE, a crystal-structure identification method based on Bayesian deep learning. As a major step forward, ARISE is robust to structural noise and can treat more than 100 crystal structures, a number that can be extended on demand. While being trained on ideal structures only, ARISE correctly characterizes strongly perturbed single- and polycrystalline systems, from both synthetic and experimental resources. The probabilistic nature of the Bayesian-deep-learning model allows to obtain principled uncertainty estimates, which are found to be correlated with crystalline order of metallic nanoparticles in electron tomography experiments. Applying unsupervised learning to the internal neural-network representations reveals grain boundaries and (unapparent) structural regions sharing easily interpretable geometrical properties. This work enables the hitherto hindered analysis of noisy atomic structural data from computations or experiments.

[1] Fritz-Haber-Institut der Max-Planck-Gesellschaft, 14195 Berlin-Dahlem, Germany. ✉email: leitherer@fhi-berlin.mpg.de

dentifying the crystal structure of a given material is important for understanding and predicting its physical properties. For instance, the hardness of industrial steel is strongly influenced by the atomic composition at grain boundaries, which has been studied in numerous theoretical and experimental investigations[1,2]. Beyond bulk materials, two- (2D) and one-dimensional (1D) systems have far-reaching technological applications, such as solar energy storage, DNA sequencing, cancer therapy, or even space exploration[3,4]. To characterize the crystal structure of a given material, one may assign a symmetry label, e.g., the space group. More generally, one may want to find the most similar structure within a list of given known systems. These so-called structural classes are identified by stoichiometry, space group, number of atoms in the unit cell, and location of the atoms in the unit cell (the Wyckoff positions).

Methods for automatic crystal-structure recognition are required to analyze the continuously growing amount of geometrical information on crystal structures, from both experimental and computational studies. Millions of crystal structures alongside calculated properties are available in large computational databases such as the Novel Materials Discovery (NOMAD) Laboratory[5], AFLOW[6], the Open Quantum Materials Database (OQMD)[7], Materials Project[8], or repositories specialized in 2D materials[9,10]. In scanning transmission electron microscopy (STEM)[11], atomic positions can be reconstructed from atomic-resolution images for specific systems, e.g., graphene[12]. Three-dimensional atomic positions are provided by atom probe tomography (APT)[13] and atomic electron tomography (AET) experiments[14]. Still, substantial levels of noise due to experimental limitations and reconstruction errors are present in atomic positions, e.g., distortions beyond a level that can be explained by a physical effect or, in case of APT, large amount of missing atoms (at least 20%, due to the limited detector efficiency)[15]. Crystal-structure recognition schemes should be able to classify a large number of structural classes (also beyond bulk materials) while at the same time being robust with respect to theoretical or experimental sources of inaccuracy and physically driven deviations from ideal crystal symmetry (e.g., vacancies or thermal vibrations). Given the large amount of data, the classification should be fully automatic and independent of the manual selection of tolerance parameters (which quantify the deviation from an ideal reference structure). Current methods are based either on space-group symmetry or local structure. For space-group-based approaches (notable examples being Spglib[16] and AFLOW-SYM[17]), the allowed symmetry operations are calculated directly from the atomic positions to infer a space group label. For local-structure-based approaches, the local atomic neighborhood of each individual atom is classified into a predefined list of reference structures. Examples of these methods are common neighbor analysis (CNA)[18], adaptive common neighbor analysis (a-CNA)[19], bond angle analysis (BAA)[20], and polyhedral template matching (PTM)[21]. Space-group approaches can treat all space groups but are sensitive to noise, while local-structure methods can be quite robust but only treat a handful of structural classes. Moreover, none of the available structure recognition schemes can recognize complex nanostructures, e.g., nanotubes.

To improve on the current state of the art, we build on recent advances in deep learning, which is a subfield of machine learning that yields ground-breaking results in many settings, e.g., image and speech recognition[22]. Previous work using machine learning and neural networks (NNs) for crystal-structure recognition[23–26] did not go beyond a handful of structural classes while showing robustness at the same time.

Here, we propose a robust, threshold-independent crystal-structure recognition framework (ARtificial-Intelligence-based Structure Evaluation, short ARISE) to classify a diverse set of 108 structural classes, comprising bulk, 2D, and 1D materials. Bayesian NNs[27,28] are used, i.e., a recently developed family of NNs that yields not only a classification but also uncertainty estimates. These estimates are principled in the sense that they approximate those of a well-known probabilistic model (the Gaussian process). This allows to quantify prediction uncertainty, but also the degree of crystalline order in a material. ARISE performance is compared with the current state of the art, and then applied to various computational and experimental atomic structures. Crystal characterization and identification of hidden patterns is performed using supervised learning (ARISE) as well as the unsupervised analysis (via clustering and dimensionality reduction) of the internal representations of ARISE.

## Results

**The input representation**. To apply machine learning to condensed-matter and materials science problems, the input coordinates, chemical species, and the lattice periodicity of a given atomic structure are mapped onto a suitable so-called descriptor. Here, the descriptor is a vector that is invariant under rigid translations and rotations of the input structure, as well as under permutations of same-species atoms. Quality and generalization ability of machine-learning models can be significantly increased, if physical requirements known to be true are respected by construction (see Supplementary Methods for more details).

Most well-known descriptors in physics and materials science incorporate these physical invariants: symmetry functions[29], the smooth-overlap-of-atomic-positions descriptor (SOAP)[30,31], the many-body tensor representation[32], and the moment tensor potential representation[33]. In this work, SOAP is used as descriptor (cf. Supplementary Methods). SOAP has been successfully applied to numerous materials science problems such as interatomic potentials fitting[34], structural similarity quantification[35], or prediction of grain boundary characteristics (e.g., energy and mobility)[36]. Note that any other suitable descriptor that respects the above-mentioned physical requirements can be used as input for our procedure. In particular, the ai4materials code library is provided into which alternative descriptors can be readily integrated.

**The Bayesian deep learning model and the training dataset**. Once the crystal structures are converted into vectorial descriptors by means of the SOAP mapping, a NN model is used to arrive at a classification decision (cf. Fig. 1c). NNs are nonlinear machine-learning models: they transform the input in a hierarchical fashion by subsequently applying affine and non-linear transformations in a predefined series of layers. The NN learns these optimal transformations that deform the descriptor space so that a robust classification is achieved. In this way, the model is able to learn complex representations which are becoming more abstract from layer to layer[26]. This ability to learn representations[37] is one of the key characteristics distinguishing NNs from other machine-learning algorithms. Various NN architectures have been developed in recent years[22]; in this work, a fully connected NN (multilayer perceptron) is employed.

A key component of this work is something rarely addressed in machine learning applied to materials science: quantification of model prediction uncertainty (cf. Fig. 1d). Standard NNs are unable to provide reliable model uncertainty[27]. In a classification setting, there is widespread use of the probability provided by the last layer as uncertainty estimate. These probabilities are typically obtained by normalizing the sum of output values using the so-called softmax activation function. The class with maximal probability corresponds to the final prediction (here of a specific structural class). One may interpret the classification probability

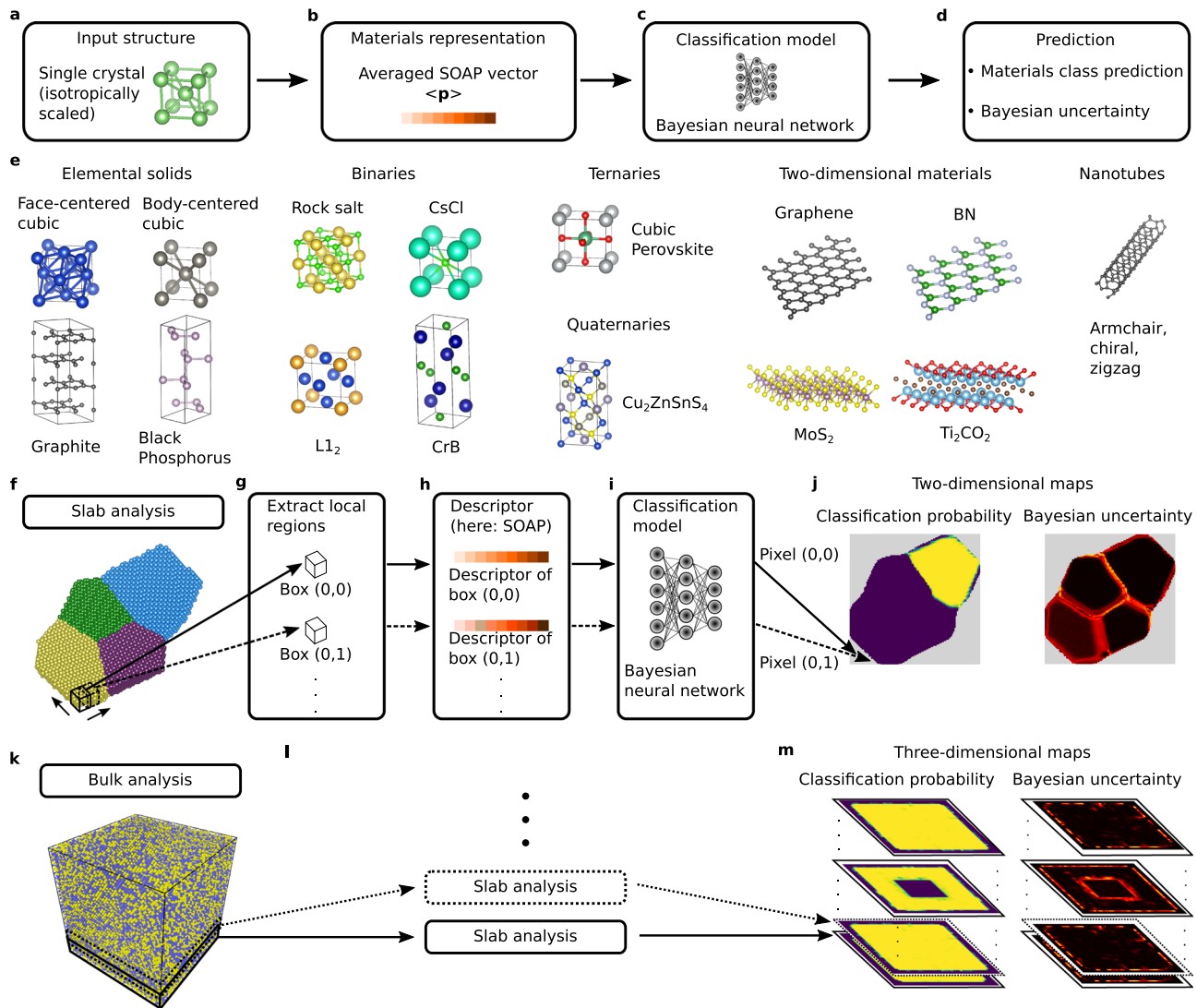

**Fig. 1 Schematic overview of single- and polycrystal characterization framework. a–d** Prediction pipeline of the single-crystal classification model ARISE (ARtificial-Intelligence-based Structure Evaluation). In this work, we employ the smooth-overlap-of-atomic-positions (SOAP) descriptor. **e** Examples of crystallographic prototypes included in the training set. **f–m** Polycrystal classification framework strided pattern matching (SPM) for slab-like (**f–j**) and bulk systems (**k–m**).

as quantification of model confidence. However, this strategy is unreliable as standard NNs tend to erroneously assign unjustified high confidence to points for which a low confidence should be returned instead[27]. The main reason for this behavior is that standard-NN predictions are deterministic, with the softmax output only providing point estimates of the true probability distribution of outputs. In Bayesian NNs, this is addressed by placing distributions over model parameters. This results in probabilistic outputs—in contrast to the point estimates from deterministic NNs—from which principled uncertainty estimates can be obtained. Gal and Ghahramani[27] showed that high-quality uncertainty estimates (alongside predictions) can be calculated at low cost using stochastic regularization techniques such as dropout[38,39] (see Supplementary Methods for more details).

After both descriptor and model architecture have been identified, a diverse, comprehensive, and materials-science-relevant training set is constructed. The first—and most important—step is to define the structural classes which are going to be included in the model: an overview of the structural classes considered in this work is shown in Fig. 1e. This comprehensive collection of structures includes bulk materials of

elemental, binary, ternary, and quaternary composition, as well as 2D materials and carbon nanotubes of chiral, armchair, and zigzag type. In practice, given any database, we extract prototypes, i.e., representative structures that are selected according to some predefined rules. Selection criteria are, for instance, fulfillment of geometrical constraints (number of atoms in the unit cell, number of chemical species) or if the structures are observed in experiment. For the elemental bulk materials, we extract from AFLOW all experimentally observed structures with up to four atoms in the primitive cell. This yields 27 elemental solids encompassing all Bravais lattices, with the exception of monoclinic and triclinic structures because of their low symmetry. Note that this selection includes not only the most common structures such as face-centered-cubic (fcc), body-centered-cubic (bcc), hexagonal-close-packed (hcp), and diamond (which cover more than 80% of the elemental solids found in nature[40]), but also double-hexagonal close-packed, graphite (hexagonal, rhombohedral, buckled), and orthorhombic systems such as black phosphorus. This goes already beyond previous work using NNs for crystal structure recognition[26], where a smaller set of elemental solids is considered. For binaries, we select the ten most

common binary compounds according to Pettifor[41], plus the L1$_2$ structure because of its technological relevance—for instance, it being the crystal structure of common precipitates in Ni-based superalloys[42]. This selection also includes non-centrosymmetric structures, i.e., structures without inversion symmetry, such as wurtzite. To challenge the classification method with an increasing number of chemical species, a small set of ternary and quaternary materials is included as a proof-of-concept. Specifically, six ternary perovskites[43] (organometal halide cubic and layered perovskites) and six quaternary chalcogenides of A$_2$BCX$_4$ type[44] are included due to their relevance in solar cells and photo-electrochemical water splitting devices, respectively. Going beyond bulk materials, we add an exhaustive set of 46 2D materials, comprising not only the well-known elemental structures such as graphene and phosphorene[45] but also binary semiconductors and insulators (BN, GaN), transition metal dichalcogenides (MoS$_2$), and one example of metal-organic

perovskites with six different chemical species. Ternary, quaternary, and 2D materials are taken from the computational materials repository (CMR)[46]. To demonstrate the ability of the proposed framework to deal with complex nanostructures, 12 nanotubes of armchair, chiral, and zigzag type are included in the dataset. For each prototype, we calculate the SOAP vector with different parameter settings (see Supplementary Methods for more details) as well as periodic and non-periodic boundary conditions to have a comprehensive dataset to train a robust classification model. This results in 39,204 (pristine) structures included in the training set.

To optimize the model, the set of pristine structures is split, with 80% being used for training and the remaining 20% for validation. For hyperparameter tuning, we employ Bayesian optimization[47], which allows to optimize functions whose evaluation is computationally costly, making it particularly attractive for deep-learning models. Here, hyperparameters such as learning rate or number of layers are optimized in an automatic, reproducible, and computationally efficient manner to minimize the validation accuracy. A list of candidate models is then obtained, from which the optimal model is selected (see "Methods" section). We term this model ARISE, and report its architecture in Table 1.

**Benchmarking**. We now compare ARISE's performance on pristine and defective structures with state-of-the-art crystal-structure recognition methods, specifically spglib, CNA, a-CNA, BAA, and PTM (cf. Table 2). As mentioned in the Introduction, none of the benchmarking methods can treat all the materials shown in Fig. 1e; thus for fairness, the classification accuracy is only calculated for classes for which the respective methods were designed for, implying that most structures are excluded (see Supplementary Note 1 for more details).

The performance on pristine structures is reported in Table 2. The accuracy in classifying pristine structures is always 100% as expected, with the only exception being CNA: For this method,

---

**Table 1 Architecture of the fully connected Bayesian neural network used in this work.**

| Layer type | Specifications |
|---|---|
| Input Layer + Dropout | Materials representation (SOAP descriptor, size: 316) |
| Fully connected layer + Dropout + ReLU | Size: 256 |
| Fully connected layer + Dropout + ReLU | Size: 512 |
| Fully connected layer +Dropout + ReLU | Size: 256 |
| Fully connected layer +Softmax | Size: 108 (= # classes) |

Rectified Linear Unit (ReLU) activation functions are used for all hidden layers. The dropout ratio is 3.17% for all layers. The total number of parameters is 371,820. While training time was fixed to 300 epochs, hyperopt found a batch size of 64 and a learning rate of $2.16 \times 10^{-4}$.

---

**Table 2 Accuracy in identifying the parent class of defective crystal structures.**

| | Pristine | Random displacements ($\delta$) | | | | | Missing atoms ($\eta$) | | | |
|---|---|---|---|---|---|---|---|---|---|---|
| | | 0.1% | 0.6% | 1% | 2% | 4% | 1% | 5% | 10% | 20% |
| Spglib, loose (96/108) | 100.00 | 100.00 | 100.00 | 95.26 | 20.00 | 0.00 | 11.23 | 0.00 | 0.00 | 0.00 |
| Spglib, tight (96/108) | 100.00 | 0.00 | 0.00 | 0.00 | 0.00 | 0.00 | 11.23 | 0.00 | 0.00 | 0.00 |
| PTM (12/108) | 100.00 | 100.00 | 100.00 | 100.00 | 100.00 | 100.00 | 88.67 | 51.76 | 25.93 | 6.24 |
| CNA (3/108) | 66.14 | 62.81 | 62.81 | 54.55 | 32.34 | 31.41 | 55.86 | 32.50 | 15.75 | 3.07 |
| a-CNA (3/108) | 100.00 | 100.00 | 100.00 | 100.00 | 100.00 | 100.00 | 89.25 | 52.81 | 25.92 | 5.37 |
| BAA (3/108) | 100.00 | 100.00 | 100.00 | 100.00 | 100.00 | 97.85 | 99.71 | 88.78 | 65.21 | 25.38 |
| GNB (108/108) | 62.63 | 56.50 | 55.94 | 55.56 | 54.98 | 52.72 | 54.51 | 52.94 | 52.67 | 52.09 |
| BNB (108/108) | 75.76 | 65.56 | 65.19 | 63.61 | 61.58 | 56.58 | 65.49 | 64.00 | 62.43 | 60.48 |
| ARISE (108/108) | 100.00 | 100.00 | 100.00 | 100.00 | 99.86 | 99.29 | 100.00 | 100.00 | 100.00 | 99.85 |

The defective structures are generated by randomly displacing atoms according to a uniform distribution on an interval $[-\delta \cdot d_{NN}, +\delta \cdot d_{NN}]$ proportional to the nearest neighbor distance $d_{NN}$ (central panel), or removing $\eta$% of the atoms (right panel). The accuracy values shown are in percentage. For benchmarking we use Spglib[16] (with two settings for the precision parameters, so-called loose (position/angle tolerance 0.1Å/5°) and tight (position/angle tolerance $10^{-4}$/1°)), polyhedral template matching (PTM)[21], common neighbor analysis (CNA)[18], adaptive common neighbor analysis (a-CNA)[19], and bond angle analysis (BAA)[20]. The number of classes which can be treated out of the materials pool in Fig. 1e is shown in parentheses for each method. spglib can assign a space group to all materials except the 12 nanotubes. PTM can only classify 7 elemental and 5 binary materials of those considered in this work. Additional classes are missing for CNA, a-CNA, and BAA as they cannot classify simple cubic (sc) and diamond structures. The approach proposed here can be applied to all classes, and thus the whole dataset is used (see Supplementary Tables 4-8 for a complete list). Moreover, we compare ARISE to a standard Bayesian approach: Naive Bayes (NB). We consider two different variants of NB: Bernoulli NB (BNB) and Gaussian NB (GNB)—see the "Methods" section for more details. ARISE is overwhelmingly more accurate than both NB methods, for both pristine and defective structures.

the default cutoff only allows to correctly classify fcc and bcc but not hcp structures. For defective structures, the situation is drastically different. Spglib classification accuracy on displaced structures is low, and only slightly improved by using loose setting (up to 1% displacement). For missing atoms, the accuracy is very low already at the 1% level regardless of the setting used. Note, however, that this is actually spglib's desired behavior since the aim of this method is not robust classification. As indicated in the first column of Table 2, spglib can treat 96 out of the 108 prototypes included in our dataset with the twelve missing prototypes being carbon nanotubes. Methods based on local atomic environments (PTM, BAA, CNA, a-CNA) perform very well on displaced structures, but they suffer from a substantial accuracy drop for missing-atoms ratios beyond 1%. Their biggest drawback, however, is that they can treat only a handful of classes: three classes for BAA, CNA, and a-CNA, and twelve classes for PTM. ARISE is very robust with respect to both displacements and missing atoms (even concurrently, cf. Supplementary Table 3), while being the only method able to treat all 108 classes included in the dataset, including complex systems, such as carbon nanotubes. An uncertainty value quantifying model confidence is also returned, which is particularly important when investigating defective structures or inputs that are far out of the training set. We provide a detailed study in Supplementary Note 3 and Supplementary Fig. 2, where we challenge ARISE with structures it has not been trained on, i.e., it is forced to fail by construction. We find that ARISE returns non-trivial physically meaningful predictions, thus making it particularly attractive, e.g., for screening large and structurally diverse databases. Moreover, we analyze predictions and uncertainty of ARISE for continuous structural transformations (cf. Supplementary Note 2 and Supplementary Fig. 1), where we consider the so-called Bain path that includes transitions between fcc, bcc, and tetragonal structures. We also want to emphasize that compared to available methods, the classification via ARISE does not require any threshold specifications (e.g., precision parameters as in spglib).

**Polycrystal classification**. Up to this point, we have discussed only the analysis of single-crystal (mono-crystalline) structures, using ARISE. To enable the local characterization of poly-crystalline systems, we introduce strided pattern matching (SPM). For slab-like systems (cf. Fig. 1f), a box of predefined size is scanned in-plane across the whole crystal with a given stride; at each step, the atomic structure contained in the box is repre-sented using a suitable descriptor (cf. Fig. 1g, h), and classified (Fig. 1i), yielding a collection of classification probabilities (here: 108) with associated uncertainties. These are arranged in 2D maps (Fig. 1j). The classification probability maps indicate how much a given polycrystalline structure locally resembles a specific crystallographic prototype. The uncertainty maps quantify the statistics of the output probability distribution (cf. Supplementary Methods). Increased uncertainty indicates that the corresponding local segment(s) deviates from the prototypes known to the model. Thus, these regions are likely to contain defects such as grain boundaries, or more generally atomic arrangements dif-ferent from the ones included in training. For bulk systems (Fig. 1k), the slab analysis depicted in Fig. 1f–j is repeated for multiple slices (Fig. 1l), resulting in 3D classification probability and uncertainty maps (Fig. 1m).

SPM extends common approaches such as labeling individual atoms with symmetry labels[19], as the striding allows to discover structural transitions within polycrystals in a smooth way. SPM can be applied to any kind of data providing atomic positions and chemical species. Results obtained via SPM are influenced by the

quality of the classification model as well as box size and stride (see "Methods" section for more details).

**Synthetic polycrystals**. First, the classification via SPM combined with ARISE is performed for a slab-like synthetic polycrystal consisting of fcc, bcc, hcp, and diamond grains (cf. Fig. 2a). Due to the nature of the system, the SPM boxes near the grain boundaries will contain mixtures of different crystal structures. The results are shown in Fig. 2b and c: The network assigns high classification probability to the correct prototypes. Uncertainty is low within the grains, increasing at grain boundaries and crystal outer borders in line with physical intuition. The result remains virtually unchanged when introducing atomic displacements (up to 1% of the nearest neighbor distance) while concurrently removing 20% of the atoms (cf. Supplementary Fig. 4). The highest classification probabilities (after from the top four shown in Fig. 2b) are shown in Supplementary Fig. 7; a discussion on the stride can be found in Supplementary Fig. 8.

Going beyond classification, we show how unsupervised learning can be used to access structural similarity information embedded in ARISE's internal representations, and use it for the characterization of crystal systems. We consider the mono-species polycrystal shown in Fig. 2a and collect ARISE's representations of the overall 7968 local boxes. Next, we employ Hierarchical Density-based Spatial Clustering Applications with Noise (HDBSCAN)[48,49] to identify clusters in the high-dimensional representation space. HDBSCAN estimates the density under-lying a given dataset and then constructs a hierarchy of clusters, from which the final clustering can be obtained via an intuitive and tunable parameter (see "Methods"). The obtained clusters correspond to the four crystalline grains in the structure (Fig. 2d). Points identified as outliers (marked in orange) coincide with grain-boundary and outer-border regions. Next, the high-dimensional manifold of the NN representations is projected in 2D via Uniform Manifold Approximation and Projection (UMAP)[50]. UMAP models the manifold underlying a given dataset and then finds a low-dimensional projection that can capture both global and local distances of the original high-dimensional data. This returns a structure-similarity map (Fig. 2e), which allows to visually investigate similarities among structures: points (structures) close to each other in this map are considered to be similar by the algorithm. Structures belonging to the same cluster are in close proximity to each other, and clearly separated from other clusters. Conversely, outlier points are split across different regions of the map. This is physically meaningful: outliers are not a cohesive cluster of similar structures, but rather comprise different types of grain boundaries (i.e., fcc to bcc transitions or fcc to diamond, etc., cf. Supplementary Fig. 9). In this synthetic setting, we can also use the classification prediction to further verify the unsupervised analysis: the results obtained via unsupervised learning indeed match ARISE's predictions (cf. Fig. 2e–f). Moreover, an analysis of the mutual information (Fig. 2g) reveals that points at the core of the clusters are associated with low uncertainty, while points closer to the boundaries show increased uncertainty. Similar results are obtained for the other layers (cf. Supplementary Fig. 6).

We now move to a more realistic system: a model structure for Ni-based superalloys[42] (cf. Fig. 2h). Ni-based superalloys are used in aircraft engines due to their large mechanical strength at high temperatures, which derives from ordered $L1_2$ precipitates ($\gamma'$ phase) embedded in a fcc matrix ($\gamma$ phase). We generate an atomic structure consisting of a fcc matrix in which Al and Ni atoms are randomly distributed. In the center, however, the arrangement of Al and Ni atoms is no longer random, but it is ordered such that the $L1_2$ phase is created (cf. Fig. 2h). The cubic

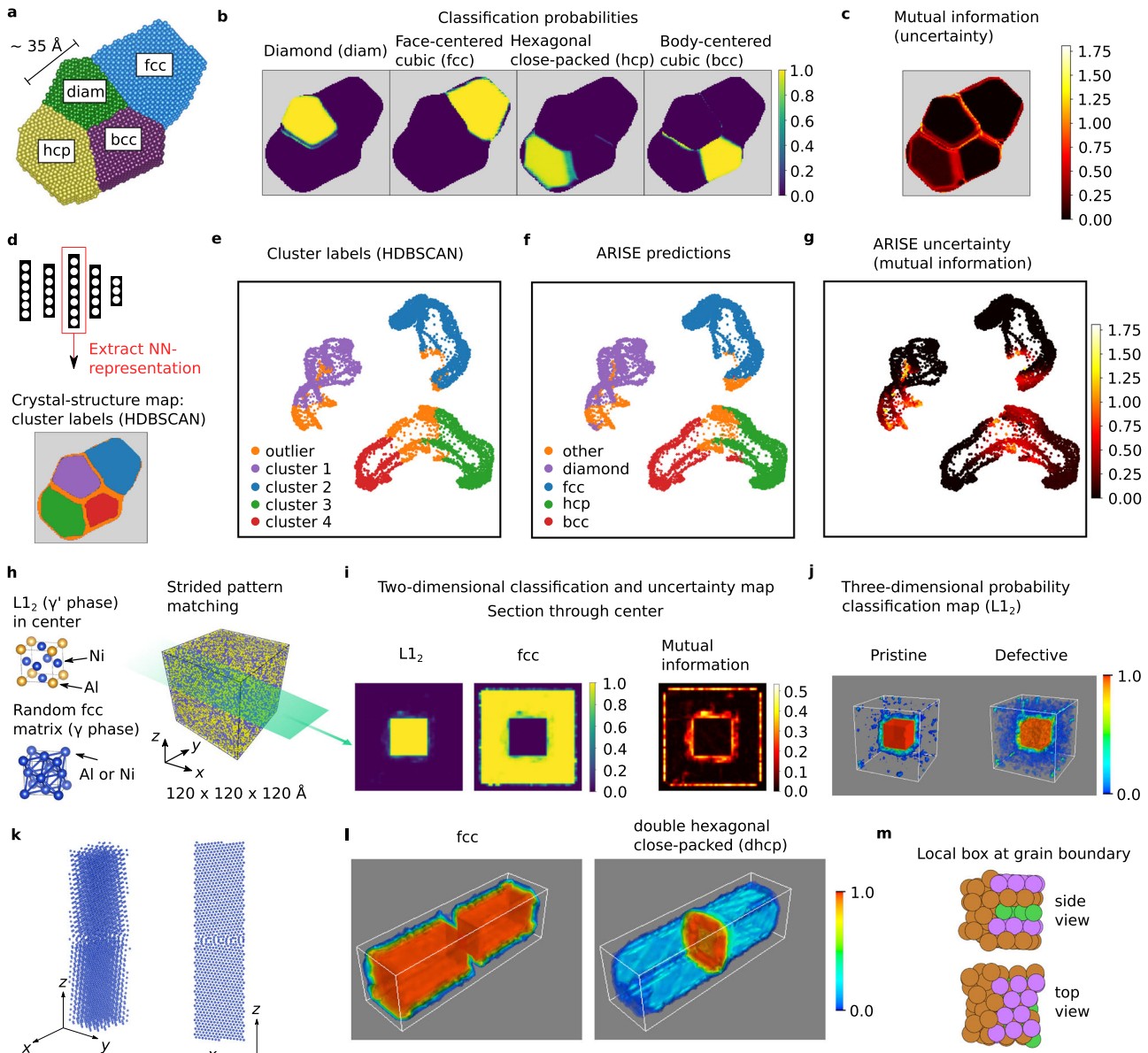

**Fig. 2 Analysis of synthetic polycrystals. a** Mono-species polycrystal consisting of four grains with face-centered cubic (fcc), body-centered cubic (bcc), hexagonal close-packed (hcp), and diamond (dia) symmetry. **b** Classification probabilities of expected prototypes. **c** Mutual information map for uncertainty quantification. **d–g** Unsupervised analysis of internal neural-network representations. **d** The neural-network representations are extracted for each local segment in (**a**) (obtained via SPM). Clustering (via Hierarchical Density-based Spatial Clustering Applications with Noise, HDBSCAN) is applied to this high-dimensional space; the polycrystal is marked according to the resulting clusters (see legend in (**e**) for the color assignments). **e–g** Two-dimensional projection (via Uniform Manifold Approximation and Projection, UMAP) of neural-network representations colored by cluster label, ARISE predicted class, and mutual information, respectively. In **e** all points for which HDBSCAN does not assign a cluster are labeled as outlier. In **f** all points that are not classified as fcc, diamond, hcp or bcc are labeled as other. Note that while the distances between points are meaningful, the axes merely serve as a bounding window and are not interpretable, a situation typically encountered in non-linear methods such as UMAP (cf. section 6[50]). **h–j** Precipitate detection in Ni-based superalloys. **h** Binary model system (right) and depiction of the two appearing phases (left). **i** Classification probabilities of expected prototypes and mutual information for a slice through the center of the structure. **j** 3D-resolved detection of the precipitate via the L1₂ classification probability for the pristine (left) and highly-defective case (right), for which 20% of the atoms are removed and randomly displaced (up to 5% of the nearest neighbor distance). **k** Lowest-energy grain boundary structure (Cu, fcc) predicted from an evolutionary search. The so-called Pearl pattern appears at the grain boundary, which is also observed in experiment[2]. **l** SPM-ARISE analysis, correctly identifying fcc (ABC close-packing) in the grains, while detecting double hexagonal close-packed (dhcp, ABAC) at the grain boundary. **m** Exemplary analysis of a local box at the grain boundary, illustrating a change in stacking and increased distortions, which motivates the assignment of dhcp (which contains 50% of both fcc and hcp close-packings).

shape of this precipitate is in accordance with experimental observations[51]. The resulting structure comprises 132 127 atoms over a cube of 120 Å length. As shown via a section through the center in Fig. 2i, fcc is correctly assigned to the matrix, and the precipitate is also detected. The uncertainty is increased at the

boundary between random matrix and precipitate, as well as at the outer borders. Figure 2j illustrates the L1₂ classification probability in a 3D plot. The precipitate is detected in both pristine and highly-defective structures. This analysis demonstrates that ARISE can distinguish between chemically ordered

and chemically disordered structures, a feature that will be exploited for the analysis of experimental data in section "Application to atomic-electron-tomography data".

Another realistic system is shown in Fig. 2k, which is the lowest-energy structure obtained from an evolutionary structure search[2]. The structural patterns at the grain boundary are also observed in scanning transmission electron microscopy (STEM) experiments. SPM-ARISE correctly identifies the fcc symmetry within the grains (cf. Fig. 2l) while assigning double hexagonal close-packed (dhcp) symmetry at the grain boundary. The local boxes at the grain boundary contain partial fcc structures while changes in stacking and distortions decrease their symmetry (cf. Fig. 2m). Also the dhcp phase (ABAC close-packing) contains fcc (ABC) and a lower-symmetry packing (hcp, AB), thus justifying the assignment. To supplement this study, we investigate several examples from the largest, currently available grain-boundary database[52], including fcc, bcc, hcp, and dhcp symmetry as well as various grain boundary types, which ARISE can classify correctly (cf. Supplementary Fig. 12). Note that ARISE correctly identifies even the $\alpha - Sm$-type stacking (ABCBCACAB). No other fully automatic approach offers a comparable sensitivity.

**Application to transmission-electron-microscopy experimental images.** We now investigate defective structures originating from a completely different data source, namely STEM experiments, to demonstrate the generalization ability of ARISE and its applicability to experimental data. Moreover, we show how global and local analysis can be combined to analyze crystal structures. STEM experiments are a valuable resource to characterize material specimens, and to study, for instance, the atomic structures at grain boundaries[2]. Atomic resolution can be achieved in high-angle annular dark field (HAADF) images. The global assignments of ARISE are tested on two experimental HAADF images of graphene shown in Fig. 3a. These images contain a substantial amount of noise which makes it very challenging to recognize the graphene honeycomb pattern by naked eye. The choice of graphene is motivated by it being a flat 2D materials; $x$ and $y$ atomic positions obtained from STEM images thus provide the actual crystal structure, and not a mere projection. Approximate atomic positions (i.e. $x$ and $y$ coordinates) from HAADF images are obtained via AtomNet[12], and shown in Fig. 3b. ARISE is then used to classify the structures following the steps summarized in Fig. 1a–d. The top predictions ranked by classification probability are shown in Fig. 3c, together with the uncertainty of the assignments as quantified by the mutual information. ARISE correctly recognizes both images as graphene, despite the substantial amount of noise present in images and reconstructed atomic positions. For the first image (Fig. 3a, left), graphene is predicted with very high probability (~99%). Indeed, the similarity to graphene is apparent, although evident distortions are present in some regions (e.g., misaligned bonds marked in Fig. 3b). The second candidate structure is $C_3N$, predicted with ~1% probability; in $C_3N$, atoms are arranged in a honeycomb lattice, making also this low probability assignment physically meaningful. For the second image (Fig. 3a, right), ARISE also correctly predicts graphene, this time with 79% probability. The uncertainty is six times larger than in the previous case. Indeed, this structure is much more defective than the previous one: it contains a grain boundary in the lower part, causing evident deviations from the pristine graphene lattice, as illustrated in Fig. 3b (right). The other four candidate structures appearing in the top five predictions (PbSe, $MnS_2$, BN, $C_3N$) are the remaining completely flat monolayers known to the network (out of the 108 structures in the training dataset, only five are flat monolayers). Note that no explicit information about the

dimensionality of the material is given to the model. It is also important to point out that ARISE robustness well beyond physical levels of noise is essential to achieve the correct classification despite the presence of the substantial amount of noise from both experiment and atomic position reconstruction.

Besides the separate classification of single images, ARISE also learns meaningful similarities between images (i.e. structures). To demonstrate this, we analyze a library of graphene images with Si defects[53] and quantify their similarity using ARISE's internal representations. Figure 3d investigates a selection of images that contain the mono-species structures of Fig. 3a (right), e, and systems with up to four Si atoms. Atomic positions are determined via AtomNet. Then, the internal representations from ARISE are extracted and the pairwise cosine similarity is calculated. The cross-similarity matrix is depicted in Fig. 3d, revealing a block matrix form in which the binary and mono-species structures are separated, i.e., more similar to each other, which can be attributed to the number of Si defects. This characteristic reappears for a larger selection of structures (cf. Supplementary Fig. 13), thus confirming the analysis of Fig. 3d. This investigation demonstrates that ARISE learns meaningful similarities, supporting the general applicability of ARISE for similarity quantification.

While so far we have analyzed HAADF images on a global scale, a local analysis via SPM allows to zoom into a given structure and locate sub-structural features. This is particularly useful for polycrystalline and/or larger systems (e.g., more than 1000 atoms). As illustrative example, we consider the structure in Fig. 3e. The mutual information shown in Fig. 3g (right) clearly reveals the presence of a grain boundary. In Fig. 3g (left), the classification probabilities of graphene and $MnS_2$ (the dominant prototypes) are presented, the latter being assigned at the grain boundary. This assignment can be traced back to pentagon-like patterns appearing near the grain boundary (as highlighted in Fig. 3e), a pattern similar to the one being formed by Mn and S atoms in $MnS_2$ (cf. Fig. 3f).

Next, we challenge the established procedure for the local analysis of 2D images with data from a completely different resource. We investigate a high-resolution transmission electron microscopy (HTREM) image of a quasicrystalline structure[54,55], cf. Fig. 3h. The bright spots are ordered aperiodically, making it a very hard task to identify the underlying order by eye. Via the established procedure, $MnS_2$ is predicted as the most similar prototype (cf. Fig. 3i). $MnS_2$ contains pentagon patterns (cf. Fig. 3f) which can also be seen in the quasicrystal (cf. zoom in Fig. 3h). This result suggests that ARISE and SPM are novel and promising tools for the classification of quasicrystalline order in automatic fashion—a promising yet under-explored area.

**Application to atomic-electron-tomography data.** While HAADF images are a valuable experimental resource, they only provide 2D projections. 3D structural and chemical information can however be obtained from atomic electron tomography (AET) with atomic resolution achieved in recent experiments[14,56–58]. Notably, this technique provides 3D atomic positions labeled by chemical species, to which ARISE and SPM can be readily applied. While extensions to other systems such as 2D materials are reported[59], metallic nanoparticles have been the main experimental focus so far, specifically FePt systems due to their promises for biomedicine and magnetic data storage[60]. First, a FePt nanoparticle[61] is classified using SPM-ARISE. ARISE's robustness is critical for this application, since the structural information provided by AET experiments are based on reconstruction algorithms that cause visible distortions (cf. Fig. 4a). SPM-ARISE primarily detects $L1_2$, $L1_0$, and fcc phases (see

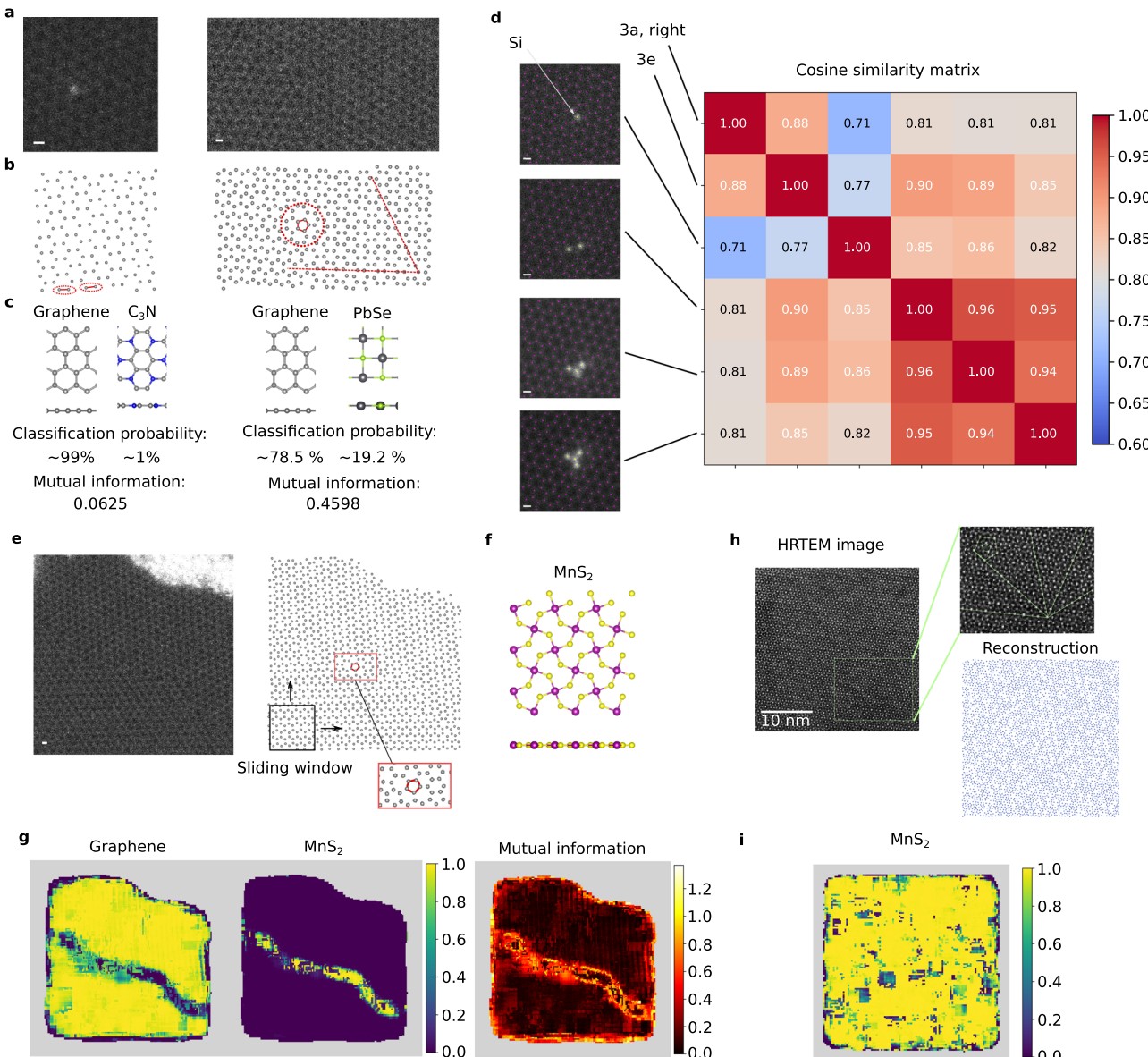

**Fig. 3 Analysis of HAADF and HRTEM images via ARISE and SPM. a** Experimental high-angle annular dark field (HAADF) images of two graphene structures. White scale bars in all HAADF images in this figure are positioned in the bottom left and correspond to the typical graphene bond length (1.42 Å). **b** The atomic positions are reconstructed from the images via AtomNet[12]. **c** The resulting atomic structures are analyzed using ARISE. The top predicted structures are shown. Mutual information is used to quantify the classification uncertainty. **d** Similarity quantification of HAADF images via ARISE. The images in **a** (right) and **e** are compared to a selection of graphene systems with Si defects[53]. For each image, AtomNet is used for reconstruction and the internal representations of ARISE are extracted (here, second hidden layer). Then, the cross-similarity is calculated using the cosine similarity. A block matrix structure arises that correlates with the number of Si atoms. A similar pattern is observed for a larger selection of structures, cf. Supplementary Fig. 13. **e** HAADF image and reconstructed atomic positions (analogous to **a**, **b**) of a larger sample. Pentagons can be spotted near the grain boundary (see inset). **f** $MnS_2$ prototype. **g** Local analysis via strided pattern matching: graphene is the dominant structure. Different prototypes ($MnS_2$) are only assigned—and with high uncertainty (mutual information)—at the grain boundary. **h** High resolution transmission electron microscopy (HTREM) image of a quasicrystalline structure (icosahedral Al-Cu-Fe, adapted from the original reference[54], see "Methods"). While there is an underlying order, the structure is aperiodic (i.e., no translational symmetry is present). As visualized in the zoom, the bright spots align with five-fold symmetry axes and pentagons of different size appear. Based on the reconstruction via AtomNet (bottom right), ARISE (via strided pattern matching) identifies $MnS_2$ as the dominating prototype (**i**), which similarly to the input structure contains pentagon patterns (**f**).

Supplementary Fig. 10). This is in line with physical expectations: annealing leads to structural transitions from chemically disordered to ordered fcc (A1 to $L1_2$) or to the tetragonal $L1_0$ phase[60,61]. Besides the expected prototypes, ARISE also finds regions similar to tetragonally distorted, mono-species fcc (cf. Supplementary Fig. 10), which is meaningful given the presence of fcc and the tetragonal phase $L1_0$.

To go beyond the information provided by classification and discover hidden patterns and trends in AET data, we conduct an exploratory analysis using unsupervised learning on ARISE's internal representations. While the procedure is similar to the one presented in Fig. 2d–g, here the analysis is truly exploratory (no ground truth is known), and data comes from experiment. First, all SPM boxes classified as $L1_0$ are extracted, this choice

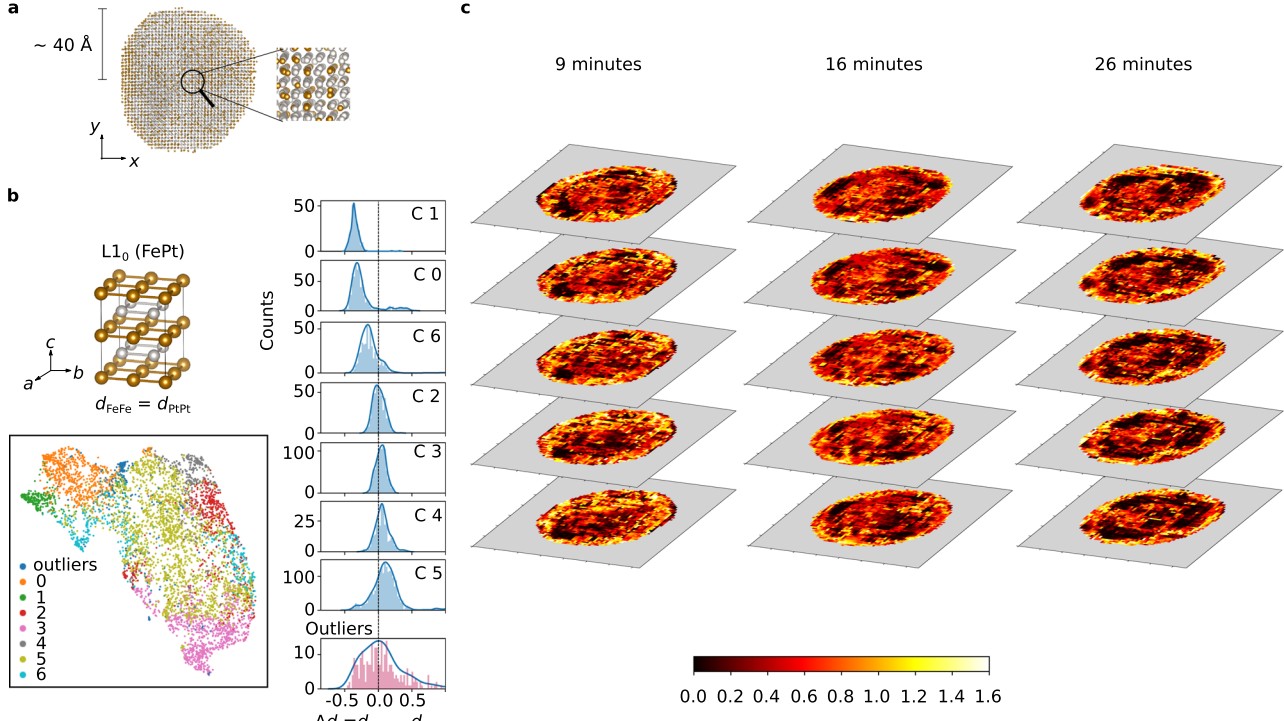

**Fig. 4 Analysis of atomic electron tomography data. a** Side view of FePt nanoparticle (~23 k atoms), with atomic positions and chemical species from atomic electron tomography (AET) data[61]. **b** Two-dimensional projection (bottom left) of neural-network representations (first hidden layer) via UMAP for regions classified as L1$_0$ by ARISE. The distribution of the difference between the nearest neighbor distances $d_{FeFe}$ and $d_{PtPt}$ (highlighted by bonds in top left part) is shown for each cluster (right), where cluster $i = 0, \ldots, 6$ is denoted as Ci, while all points for which HDBSCAN does not assign a cluster are labeled as outlier. **c** Five central slices (mutual information, obtained via strided pattern matching) for three different annealing times (data from four-dimensional AET experiment[62]).

motivated by the physical relevance of this phase, in particular, due to its magnetic properties[60]. This reduces the number of data points (boxes) from 43,679 to 5359—a significant filtering step for which the automatic nature of ARISE is essential. In the representation space of the first hidden layer, HDBSCAN identifies seven clusters (and the outliers). To interpret the cluster assignments, we analyze geometrical characteristics of atomic structures (i.e., the local boxes) assigned to the different clusters. Specifically, we consider the nearest neighbor distances between Fe and Pt atoms, $d_{FeFe}$ and $d_{PtPt}$, respectively (cf. Supplementary Methods for the definition). For an ideal tetragonal structure, the difference $\Delta d = d_{FeFe} - d_{PtPt}$ is zero (cf. Fig. 4b, top left); a deviation from this value thus quantifies the level of distortion. Looking at the histograms of the (signed) quantity $\Delta d$ shown in Fig. 4b for each cluster, one can observe that each distribution is peaked; moreover, the distribution centers vary from negative to positive $\Delta d$ values across different clusters. The distribution of the outliers is shown for comparison: the $\Delta d$ distribution is very broad, since outlier points are not a meaningful cluster. While overlap exists, the clusters correspond to subgroups of structures, each distorted in a different way, as quantified by $\Delta d$. Thus, we discovered a clear trend via the cluster assignment that correlates with the level of distortion. The cluster separation can be visualized in 2D via UMAP (cf. Fig. 4b). Notably, the clusters do not overlap, even in this highly compressed representation (from 256 to 2 dimensions). Some of the clusters may also contain further sub-distributions, which seems apparent for instance from the $\Delta d$ distribution of cluster 6. The regions corresponding to the clusters could be hinting at a specific growth mechanism of the L1$_0$ phase during annealing, although further investigations are necessary to support this claim. The present analysis provides a protocol for the machine-learning driven exploration of structural data: supervised learning is employed to filter out a class of interest (which is not a necessary step, cf. Fig. 2d–g), then unsupervised learning is applied to the NN representations, revealing regions sharing physically meaningful geometrical characteristics.

Finally, we apply ARISE to time-resolved (i.e., four-dimensional) AET data. Specifically, a nanoparticle measured for three different annealing times is investigated[62]. The mutual information as obtained via SPM-ARISE is shown in Fig. 4c for five central slices. In regions between outer shell and inner core, the mutual information clearly decreases for larger annealing times, indicating that crystalline order increases inside the nanoparticle (see also Supplementary Fig. 11 for more details). This analysis confirms that the predictive uncertainty of ARISE, as quantified by the mutual information, directly correlates with crystalline order. The mutual information can be therefore considered an AI-based order parameter, which we anticipate to be useful in future nucleation dynamics studies.

## Discussion

In this work, Bayesian deep learning is employed to achieve a flexible, robust, and threshold-independent crystal classification model, which we term ARISE. This approach correctly classifies a comprehensive and diverse set of crystal structures from computations and experiments, including polycrystalline systems (via strided pattern matching). Given an unknown structure, the network assigns—in an automatic fashion—the most similar prototypes among 108 possible classes (and quantifies the similarity!), which is a very complicated task even for trained materials scientists, in particular in case of complex and possibly defective 3D structures. ARISE is trained on ideal synthetic systems only and correctly identifies crystal structures in STEM and

AET experiments, hence demonstrating strong generalization capabilities. The Bayesian deep-learning model provides classification probabilities, which—at variance with standard NNs—allow for the quantification of predictive uncertainty via mutual information. The mutual information is found to directly correlate with the degree of crystalline order, as shown by the analysis of time-resolved data from AET experiments. This demonstrates the correlation of an information-theory concept with physical intuition. The internal NN representations are analyzed via state-of-the-art unsupervised learning. The clusters identified in this high-dimensional internal space allow to uncover physically meaningful structural regions. These can be grain boundaries, but also unexpected substructures sharing geometrical properties, as shown for metallic nanoparticles from AET experiments. This illustrates how supervised and unsupervised machine learning can be combined to discover hidden patterns in materials science data. In particular, the physical content learned by the NN model is explained by means of unsupervised learning. Since ARISE is not limited to predicting the space group, systems where the space group does not characterize the crystal structure can be tackled (as demonstrated for carbon nanotubes). More complex systems such as quasi-crystals[55], periodic knots, or weavings[63] could also be considered. Indeed, ARISE can be applied to any data providing Cartesian coordinates labeled by chemical species. Practically, one simply needs to add the new structures of interest to the training set, and re-train or fine-tune (i.e., via transfer learning) the NN with the desired labels. Moreover, the mutual information allows to quantify the defectiveness of a structure; this could be exploited to automatically evaluate the quality of STEM images, for example, one may automatically screen for STEM images that are likely to contain structural defects. Applications in active learning[64] for materials science are also envisioned, where uncertainty is crucial for example, when deciding on the inclusion of additional—typically computationally costly—points in the dataset.

## Methods

**Dataset creation**. To compute the training set (39,204 data points in total), we include periodic and non-periodic systems. For the former, no supercells are necessary (as SOAP is supercell-invariant for periodic structures). For the latter, a given structure (or rather its unit cell as obtained from the respective database) is isotropically replicated until at least 100 atoms are contained in the structure. Then this supercell structure and the next two larger isotropic replicas are included. With this choice of system sizes, we focus on slab- and bulk-like systems. Note that the network may not generalize to non-periodic structures outside the chosen supercell range. Practically, if the need to classify much smaller or larger supercells arises, one can include additional replicas to the training set and retrain the model (while for larger supercells it is expected that the network will generalize, see also Supplementary Fig. 5). Retraining is computationally easy due to fast convergence time. Note that for 2D structures, only in-plane replicas are considered.

Elemental solids and binary compounds are selected from the AFLOW library of crystallographic prototypes[6]. Ternary, quaternary, and 2D materials are taken from the computational materials repository (CMR)[46].

Nanotubes are created using the atomic simulation environment (ASE)[65] where the chiral numbers (n,m) provide the class labels. We filter out chiral indices $(n, m)$ (with the integer values $n$, $m$ taking values in [0, 10]) for which the diameter is in the range [4 Å, 6 Å] (and skip the cases where $n = m = 0$, $n < m$). Then, we increase the length of each nanotube until at least 100 atoms are contained. No additional lengths are included as it was checked that there is no major change in the SOAP descriptor (via calculating the cosine similarity between descriptors representing nanotubes of different length). For more complex nanotubes (for instance, multi-walled systems), this may change.

For the cutoff $R_C$, we select the range $[3.0 \cdot d_{NN}, 5.0 \cdot d_{NN}]$ in steps of $0.2 \cdot d_{NN}$ and for $\sigma$ the values $[0.08 \cdot d_{NN}, 0.1 \cdot d_{NN}, 0.12 \cdot d_{NN}]$. We calculate the SOAP descriptor using the quippy package (https://libatoms.github.io/QUIP), where we choose $n_{max} = 9$ and $l_{max} = 6$ as limits for the basis set expansion, resulting in an averaged SOAP vector of length 316. Furthermore, we increase the dataset by varying the so-called extrinsic scaling factor: For a given prototype, the value of $d_{NN}$ will deviate from the pristine value in presence of defects. Thus, to inform the network that the computation of $d_{NN}$ is erroneous, we scale each pristine prototype not only by $1.0 \cdot d_{NN}$ but also $0.95 \cdot d_{NN}$ and $1.05 \cdot d_{NN}$. We term the factors 0.95,

1.0, 1.05 extrinsic scaling factors. One may also see this procedure as a way to increase the training set.

To create defective structures, we explained in the main text (cf. Table 2) how defects (displacements, missing atoms) are introduced. Note that we use the term missing atoms and not vacancies since the percentages of removed atoms we consider are well beyond regimes in real materials. Also note that displacements as high as 4% of the nearest neighbor distance might already cause a transition to the liquid phase in some solids. Still, as noted in the Introduction, experimental and computational data often present levels of distortions that are comparable or even substantially exceed these regimes. We introduce defects for all pristine prototypes included in the training set (specifically, for the supercells—for both periodic and non-periodic boundary conditions, while for nanotubes only non-periodic structures are used). Since the defects are introduced randomly, we run 10 iterations of defect creation on each prototype. Then we calculate SOAP for all of these defective structures for one specific parameter setting ($R_C = 4.0 \cdot d_{NN}$, $\sigma = 0.1 \cdot d_{NN}$, extrinsic scaling factor = 1.0), which corresponds to the center of the respective parameter ranges included in the training set. Finally, we obtain 5880 defective structures for each defect ratio. In total, we compute defectives structures for three defect types (missing atoms and displacements introduced both separately and combined) for eight different defect ratios, giving in total 141,120 defective data points.

**Neural-network architecture and training procedure**. At prediction time, we need to fix $T$, the number of forward-passes being averaged (cf. Supplementary Methods). We chose $T = 10^3$ for all results except Fig. 3c and Supplementary Fig. 2, for which we increase $T$ to $10^5$ in order to get stable assignments in case of high uncertainty and very low probability candidates (i.e., <1.0%). Still, the most similar prototypes can be obtained already with $10^3$ iterations.

Training is performed using Adam optimization[66]. The multilayer perceptron is implemented in Keras[67] using Tensorflow[68] as backend. Furthermore, we optimize hyperparameters such as the number of layers using Bayesian optimization, specifically the Tree-structured Parzen estimator (TPE) algorithm as provided by the python library hyperopt[47] (cf. Supplementary Methods for more details).

The initial training set is split (80/20% training/validation split of pristine structures, performed using scikit-learn, in stratified fashion, using a random state of 42) and the accuracy on the validation set is used as the performance metric to be minimized via hyperopt (for 50 iterations). Fast convergence (followed by oscillations around high accuracy values) or divergence is typically observed, which is why we train for a fixed number of epochs (300) and save only the model with the best performance on the validation set. Training is performed on 1 GPU (Tesla Volta V100 32GB) on the Talos machine-learning cluster in collaboration with the Max Planck Computing and Data facility (MPCDF). We observe that accuracies around 99% can be reached after few iterations, with individual training runs converging within 20 min, depending on model complexity.

Practically, strong models are obtained via this procedure, while further fine-tuning can be made to reach perfect accuracies. First, we restrict to one setting of training parameters (see the previous section). From a computational efficiency point of view, this is also the preferred choice since one has to compute only one descriptor per structure during prediction time. We select $R_C = 4.0 \cdot d_{NN}$ and $\sigma = 0.1 \cdot d_{NN}$ as well as an extrinsic scaling factor of 1.0. These choices are at the center of the respective parameter ranges. While the model with highest validation accuracy (on the whole training set) determined via hyperopt usually gives very strong performance, it is not necessarily the best possible one, especially in terms of generalization ability to defective structures. To find the optimal (i.e., most robust) model we select some of the best models (e.g., top 15) found via hyperopt and rank them based on their performance on pristine and defective structures (again for one setting of $R_C$, $\sigma$). In particular, we restrict to defective points with either ≤5% atoms missing or <1% atomic displacement, which comprises 35,280 data points (six different defect ratios with 5880 points each). The number of pristine data points is 396. Using this strategy, we can identify a model with 100% accuracy on pristine and defective structures, which is reported in the last line of Table 2. The accuracy on the whole training set comprising 39,204 data points is 99.66%.

We also investigate the performance on higher defect ratios beyond physically reasonable perturbations, since this is typically encountered in atom-probe experiments. In particular, we investigate three defect types (missing atoms, displacements, and both of them) comprising 105,840 data points. The results for missing atoms (>5%) and displacements (>0.6%) can be found in Table 2 and Supplementary Table 2. Classification accuracies on structures with both missing atoms and displacements are specified in Supplementary Table 3. Note that training and model selection only on pristine structures can yield robust models, especially if the number of classes is reduced. For instance, training only on binary systems using a pristine set of 4356 data points (full SOAP parameter range) gives perfect accuracy on both the full training set and 3960 defective structures (displacements ≤0.06% and ≤5% missing atoms—for the setting $R_C = 4.0 \cdot d_{NN}$, $\sigma = 0.1 \cdot d_{NN}$, extrinsic scaling factor 1.0). Note that in general, if fewer classes are considered (e.g., ~20), the training time can be significantly reduced (e.g., to a few minutes).

**Naive Bayes**. We employ the implementation provided by scikit-learn (https://scikit-learn.org/stable/modules/naive_bayes.html), where two assumptions for the

likelihood $P(x_i|y)$ of the features $x_i$ given the labels $y$ are tested: a Gaussian distribution (Gaussian Naive Bayes, short GNB) and a multivariate Bernoulli distribution (Bernoulli Naive Bayes, short BNB). We observe that the BNB model yields improved results compared to GNB, while both being significantly less accurate than ARISE.

**Unsupervised learning: clustering and dimensionality reduction.** HDBSCAN[48,49] is a density-based, hierarchical clustering algorithm (see also the online documentation https://hdbscan.readthedocs.io/en/latest/). The final (so-called flat) clustering is derived from a hierarchy of clusters. The most influential parameter is the minimum cluster size that determines the minimum number of data points a cluster has to contain – otherwise it will be considered an outlier, i.e., not being part of any cluster. Practically, one can test a range of values for the minimum cluster size, in particular very small, intermediate, and large ones—for instance for the results on the synthetic polycrystal in Fig. 2a, we test the values {25, 50, 100, 250, 500, 1000}. In line with intuition, the number of clusters grows (shrinks) for smaller (larger) values of minimum cluster size. A coherent picture with 4 clusters and clear boundaries (as indicated by the outliers) arises for minimum cluster size values of around 500, for which we report the results in Fig. 2d–g and Supplementary Fig. 6. Moreover, we test the influence of the so-called minimum distance parameter in Supplementary Fig. 9, where for Fig. 2e–g, we choose a minimum distance parameter of 0.9.

For the nanoparticle data discussed in Fig. 4c, we observe that most of the points are considered outliers since the data contains substantially more distortions. To address this, we use the soft clustering feature of HDBSCAN, which allows to calculate a vector for each data point whose $i$-th component quantifies the probability that the given data point is member of cluster $i$. Then, we can infer a cluster assignment for points that would normally be considered outliers, by selecting for each point the cluster whose membership probability is maximal (while considering a point an outlier if all probabilities are below a certain threshold for which we choose 10%). For the minimum cluster size, we find that for values below 10 the number of clusters quickly grows while shrinking for larger values. We report the results for a minimum cluster size of 10 and a minimum distance parameter of 0.1 in Fig. 4c.

To visualize the clustering results, we use the manifold-learning technique UMAP[50] (see also the online documentation https://umap-learn.readthedocs.io/en/latest/). This method uses techniques from Riemannian geometry and algebraic topology to capture both the global and local structure of a manifold that underlies a given dataset. One of the most important parameters is the number of neighbors that will be considered to construct a topological representation of the data, where a small value takes only the local structure into account, while a large value considers the global relations between data points. We choose values of 500 for Fig. 2e–g and 50 for 4c, above which the 2D embeddings do not change significantly.

**Synthetic polycrystal generation.** The structure in Fig. 2a is generated via the open-source software Atomsk[69].

**Strided pattern matching parameters.** Two parameters are most important for strided pattern matching analysis: firstly, the stride defines the resolution and may be chosen arbitrarily small or large to increase or decrease the visualization of structural features. Note that the sliding allows us to discover smooth transitions, while the smoothness is determined by the step size. This way, boundary effects between neighbored local regions are reduced compared to the case of slightly or non-overlapping boxes (e.g., in the simple voxelization case). In particular, a small stride (e.g., 1 Å) mitigates boundary effects due to the discretization, which otherwise can influence the final classification and uncertainty maps. SPM is trivially parallel by construction, thus allowing the time-efficient characterization of large systems. Clearly, in a naive implementation, this procedure scales cubically with stride size. Practically, one may choose a large stride (in particular if the structure size would exceed computing capabilities) to obtain low-resolution classification maps, which may suffice to identify regions of interest. Then, one may zoom into these areas and increase the stride to obtain high resolution classification maps revealing more intricate features. Secondly, the box size determines the locality, i.e., the amount of structure that is averaged to infer the crystallographic prototype being most similar to a given local region. If this parameter is chosen too large, possibly interesting local features may be smoothed out. We recommend to use box sizes larger than 10–12 Å, as in these cases, the number of contained atoms is typically within the range of the supercells the network is trained on (i.e., at least 100 atoms). The generalization ability to smaller structures depends on the prototype (cf. Supplementary Fig. 5), and in general, if a smaller box size is desired while using our model, the practical solution is to add smaller supercells in the training set and retrain the network. Note that the shape of the local regions may be chosen to be different from boxes, e.g., spheres or any other shape that fits the application at hand. Moreover, we chose the grid in which the structure is strided to be cubic, while other discretizations are possible. Note that a one-dimensional striding can be applied to rod-like systems such as carbon nanotubes.

In this work, we choose the following SPM parameters: For the slab analysis in Fig. 2a, we choose a 1 Å stride and a box size equal to the slab thickness (16 Å). For the superalloy model system we choose the same box size but reduce the stride to 3 Å, since this system is much larger and we want to demonstrate that for these systems, smaller strides still yield reasonable results. For the grain-boundary structure in Fig. 2k, a stride of 2 Å and a box size of 10 Å suffice to characterize the system. For the 2D STEM analysis (cf. Fig. 3g), we choose a stride of 4 (in units of pixels since atoms are reconstructed from images, while for typical graphene bond lengths of 1.42 Å the relation 1 Å ≈ 8.5 can be inferred). Moreover, we select a box size of 100 pixels (≈12 Å). For the quasicrystalline structure in Fig. 3h, i, which has been cropped from the original reference[54] and rescaled to a $1000 \times 1000$ pixel image (using standard settings in the GIMP Image editor), a box size of 100 pixels and stride of 10 pixels suffice to detect the $MnS_2$ prototype. For the nanoparticle analysis, we choose a stride of 1 Å and box size of 12 Å for all of Fig. 4, except the clustering analysis, for which we reduce the stride to 2 Å, to avoid an overcrowded 2D map. The box size of 16 Å (which allowed to distinguish chemically disordered fcc from ordered $L1_2$, cf. Fig. 2h–j) yields comparable results (see Supplementary Fig. 10), while finding less $L1_0$ symmetry and more fcc since a larger amount of structure is averaged. Due to $L1_0$ showing special magnetic properties, we are interested in having a larger pool of candidate regions, which is why we choose a box size of 12 Å (corresponding to the smallest value such that the average number of atoms in each box is greater than 100).

**Atomic electron tomography.** ARISE's predictions are reliable since all the symmetries that typically occur in FePt nanoparticles are included in the training set—except the disordered phase for which it has been demonstrated in the analysis of the Ni-based superalloy model system that ARISE is sensitive to chemical ordering. Moreover, a supplementing study reveals that ARISE can analyze structural transformations, in particular similar to the ones taking place in nanoparticles (cf. Supplementary Note 2 and Supplementary Fig. 1, where the so-called Bain path is investigated).

Due to diffusion, the shape of the three nanoparticles (cf. Fig. 4c) and thus the number of atoms is changing. Rough alignment of the nanoparticles was checked using point set registration: Specifically, we employed the coherent point drift algorithm[70] as implemented in the python package pycpd (https://github.com/siavashk/pycpd). We extracted only the core of the nanoparticle, which is reported to remain similar during the annealing procedure[14]. After applying the algorithm, the remaining mismatch is negligible (3–10° for all three Euler angles).

## Data availability
The training and test data, trained neural-network model, as well as all relevant geometry files and datasets that are generated in this study have been deposited at Zenodo under accession code https://doi.org/10.5281/zenodo.5526927. The geometry file of the so-called Pearl structure analyzed in Fig. 2k–m is available in Edmond (the Open Access Data Repository of the Max Planck Society) under accession code https://edmond.mpdl.mpg.de/imeji/collection/zV4i2cu2bIAI8B. The experimental HAADF image datasets and trained neural-network models that are employed in this study for reconstructing atomic positions are available under accession codes https://github.com/pycroscopy/AICrystallographer/tree/master/AtomNet and https://github.com/pycroscopy/AICrystallographer/tree/master/DefectNet. The HRTEM data used in this study (Fig. 3h) has been adapted (see "Methods") from the original publication[54], where it is published under a Creative Commons Attribution 4.0 International License. The AET data used in this study is available in the Materials Data Bank (MDB) under accession code https://www.materialsdatabank.org/.

## Code availability
A Python code library ai4materials containing all the code used in this work is available at https://github.com/angeloziletti/ai4materials. In more detail, ai4materials provides tools to perform complex analysis of materials science data using machine learning techniques. Furthermore, functions for pre-processing, saving, and loading of materials science data are provided, with the goal to ease traceability, reproducibility, and prototyping of new models. An online tutorial to reproduce the main results presented in this work can be found in the NOMAD Analytics-Toolkit at https://analytics-toolkit.nomad-coe.eu/tutorial-ARISE.

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

## Acknowledgements

We acknowledge funding from BiGmax, the Max Planck Society's Research Network on Big-Data-Driven Materials Science. L.M.G. acknowledges funding from the European Union's Horizon 2020 research and innovation program, under grant agreements No. 951786 (NOMAD-CoE) and No. 740233 (TEC1p). Furthermore, the authors acknowledge the Max Planck Computing and Data Facility (MPCDF) for computational resources and support, which enabled neural-network training on 1 GPU (Tesla Volta V100 32GB) on the Talos machine learning cluster. The authors thank Matthias Scheffler for initiating this research direction and providing comments to the manuscript.

## Author contributions

All authors designed the project. A.Z. and A.L. wrote the code, A.L. performed all the calculations. A.Z. and L.M.G. supervised the project. All authors wrote and reviewed the manuscript.

## Funding

## Competing interests

The authors declare no competing interests.
