## [Peer Review File · Nature Communications]

Robust recognition and exploratory analysis of crystal structures via Bayesian deep learningEditorial Note: Parts of this Peer Review File have been redacted as indicated to remove third-party material where no permission to publish could be obtained.

REVIEWER COMMENTS

Reviewer #1 (Remarks to the Author):

This article introduces a new classification scheme for local crystal structures based on Bayesian neural networks. The proposed strategy is based on classification of local crystalline environments via the well-known SOAP descriptor. This method offers two key features distinguishing it from other neural-network-based classifiers. First, the use of Bayesian NNs provides a principled uncertainty estimate for the output class. Second, the use of strided pattern matching offers a viable scheme for scaling up the analysis to very large samples. The approach is demonstrated on several interesting and relevant case studies, including data originating from both simulation and experiment.

This work is relevant and well-executed. The selected machine learning methods are appropriate for the task at hand and appear to be performed with care. The methods (including architecture and hyperparameters) are well documented. In my opinion, the most significant result is the robustness of the method to missing atoms, which is a major problem with most existing techniques. Overall, this should be a welcome contribution to the field of crystal classification methods.

I have a few comments which should be addressed prior to publication:

(1) The authors mention many other descriptors before identifying SOAP as the one used here for the input feature vector to the NN. Can the authors comment whether they tried other descriptors and selected SOAP for superior performance, or simply selected SOAP from the beginning for some personal preference? I recognize that a systematic study by different descriptors is out of scope for this work, but I think this is a major point of interest for the community and I would very much like to know the authors' perspective on these different descriptors if they have experimented with them during this work (or even in other published works).

(2) All the UMAP results (Fig.2, Fig.S6, Fig.S9) look very tightly clustered into structures with apparently meaningless shapes. The number of neighbors is discussed in the methods section, but I wonder if increasing the minimum distance parameter would provide a smoother manifold resulting in a more interpretable topology. In effect, would the topology of the clusters associated with each pristine crystal structures converge with the topology from the sample? This could provide even stronger evidence that the learned representations are physically meaningful.

(3) p.3, Fig.1j: I find the use of a grid overlay misleading when combined with the nearby annotation of "Pixel" – at first glance, it looks like the pixels are each grid cell. These are omitted from all later SPM-based images, so I suggest they be omitted here as well.

Finally, a few typos I noticed:

(4) p.6, second paragraph: order of words is mixed up in "high-angle annular dark field" (HAADF)

(5) p.8, last paragraph: "obtained via SPM is shown in Fig. 4d" should read "Fig. 4c"

Sincerely,
Wesley F. Reinhart

Reviewer #2 (Remarks to the Author):

This manuscript presents a neural network based approach to assign crystal structure to spatial data. The paper is well written and addresses an important issue. My primary concern with this manuscript is that I'm not sure that the authors have adequately shown the added benefit from this work. The following are my specific comments :

- The examples shown are either examples where the result could be seen by eye or else where the result is not proven. For example, the simulated data in Figure 2 could have more uncertainty and larger amounts of inconsistency in chemistry or with much more missing data. Then there could be a correct label but with perhaps enough variance that it can't be separated out by eye. Or along this line, the analysis in Figure 3 seems to return what can be seen by eye, and actually AtomNet seems to be doing most of the heavy lifting. And then in Figure 4, the uncertainty change

with increased annealing temperature seems obvious as the data will change (particularly as it is tomographically derived) and so I am not sure if that is sufficient for validating the result.

I may be incorrect in what I see as the information gain, and if so that should be clarified. If there was an example shown where it is highlighted what is gained that we didn't have before or some acceleration, etc. that would help emphasize the impact of this work.

- In the AET case, it is a little confusing in what the analysis is picking up. It seems the objective of the approach is to define the crystal structures but here as far as I can tell from what is written is that it is capturing orientation and the structure is all the same. This should be clarified as to what is captured here versus the extent of the approach.

- Is there any issues when looking at reconstructed, ie. aet, data as opposed to microscopy data?

- It is a different technique, but is mentioned in the introduction: in the standard APT analysis software (such as through IVAS), the bond distributions are a standard input as well as different clusters and some approximation of structure. Mention of what is added here compared to that would be helpful.

- The term 'deep learning' is used a lot now, and I think justification of how this is deep learning and not machine learning (especially since the neural network used isn't convolutional neural network) would be good.

- On page 4, start of the first paragraph, it is set that 80% of data is used for training and the rest for test. Is this randomly selected and evenly distributed? Also, in experimental data we often have large regions where the data is not evenly represented. How does the result change if the training is biased to specific regions?

Reviewer #3 (Remarks to the Author):

The manuscript presents a method that employs neural networks to classify atomic structures. Trained on a synthetic database where the entries were labelled, the authors present a number of benchmark tests where the model is capable of accurately predicting the appropriate label. Applications on experimental results are also shown.

The topic is interesting and the work is solid, but overall it is let down by the poorly written text, especially in the first half of the paper. I list a couple of remarks and examples of imprecise prose.

- The first paragraph of the introduction is somewhat difficult to comprehend.

- "NNs are non-linear classifiers": not true, they can be used as classifiers as well as regressors etc.

- "distortions beyond the physical level": I am not even sure what a distortion at the physical level may be

- "Standard NNs are unable to provide reliable model classification uncertainty. There is a widespread use of the probability provided by the last layer as uncertainty estimate. Typically, the so-called softmax function is employed to normalize the sum of output values such that they can be interpreted as classification probabilities.": how do these sentences belong together?

Regarding the method, one may ask if using a neural network is really necessary. There are numerous Bayesian classifier methods which do not use NNs, it would interesting to know whether these were considered but rejected or NNs were chosen mainly for the hype factor. Otherwise the methodology is based on some well-established tools, combined with materials databases.

Considering the applications, I found those presented in Fig. 2 indeed synthetic, as these were not even applied on results of atomic simulations but constructed for demonstrations. I would have been nice to see more realistic scenarios covered here. In the interpretation and classification of experimental data, these are essentially 2D problems and I would be surprised if the standard image recognition machinery hasn't been applied to them. It would be interesting to see how

ARISE extracts more information and provides new understanding.

Overall, the paper in its current form cannot be accepted for publication in Nature Communications. The text needs attention and the results strengthening. Following a major revision, it should be revisited by the reviewers to see if ARISE indeed delivers.

Reviewer #1 (Remarks to the Author):

This article introduces a new classification scheme for local crystal structures based on Bayesian neural networks. The proposed strategy is based on classification of local crystalline environments via the well-known SOAP descriptor. This method offers two key features distinguishing it from other neural-network-based classifiers. First, the use of Bayesian NNs provides a principled uncertainty estimate for the output class. Second, the use of strided pattern matching offers a viable scheme for scaling up the analysis to very large samples. The approach is demonstrated on several interesting and relevant case studies, including data originating from both simulation and experiment.

This work is relevant and well-executed. The selected machine learning methods are appropriate for the task at hand and appear to be performed with care. The methods (including architecture and hyperparameters) are well documented. In my opinion, the most significant result is the robustness of the method to missing atoms, which is a major problem with most existing techniques. Overall, this should be a welcome contribution to the field of crystal classification methods.

We appreciate the referee's assessment of our work, in particular that it is relevant and well-executed.

I have a few comments which should be addressed prior to publication:

(1) The authors mention many other descriptors before identifying SOAP as the one used here for the input feature vector to the NN. Can the authors comment whether they tried other descriptors and selected SOAP for superior performance, or simply selected SOAP from the beginning for some personal preference? I recognize that a systematic study by different descriptors is out of scope for this work, but I think this is a major point of interest for the community and I would very much like to know the authors' perspective on these different descriptors if they have experimented with them during this work (or even in other published works).

(2) All the UMAP results (Fig.2, Fig.S6, Fig.S9) look very tightly clustered into structures with apparently meaningless shapes. The number of neighbors is discussed in the methods section, but I wonder if increasing the minimum distance parameter would provide a smoother manifold resulting in a more interpretable topology. In effect, would the topology of the clusters associated with each pristine crystal structures converge with the topology from the sample? This could provide even stronger evidence that the learned representations are physically meaningful.

(3) p.3, Fig.1j: I find the use of a grid overlay misleading when combined with the nearby annotation of "Pixel" – at first glance, it looks like the pixels are each grid cell. These are omitted from all later SPM-based images, so I suggest they be omitted here as well.

1.1) We agree that the choice and comparison of different descriptors is of high relevance. This has been and is topic of several projects in the NOMAD laboratory (see for instance L. M. Ghiringhelli *et al.* *Physical Review Letters* **114**, 105503, 2015). Closer to the present work, a novel diffraction fingerprint has been introduced in Ziletti *et al.* *Nature Communications* **9**, 1–10, 2018. While in principle one could adapt this descriptor for other applications, it has been employed for crystal-structure classification (using a convolutional neural network as classification model). While demonstrating a high level of robustness, this approach is restricted to a limited number of structural classes by construction, since diffraction fingerprints of non-centrosymmetric structures are indistinguishable. One notable advantage of this approach is that interpretability can be achieved since one can use so-called attentive response maps to determine which peaks in the diffraction fingerprint are used by the network to classify a given structure (while in the current manuscript, the SOAP components lack this kind of interpretability). Extending the approach to a more diverse set of structural classes, while maintaining the robustness with respect to missing and atomic

displacements, is one of the main results of the presented approach. Moreover, principled uncertainty estimates and the classification in both global and local fashion have been added. Since it is one of the state-of-the-art descriptors, we tested SOAP, while not assuming it to be sufficient and developing several extensions in parallel. For instance, we are investigating possible advancements via Fourier-transforms, a version of SOAP which we termed FT-SOAP (unpublished). The transition from real space (SOAP) to Fourier space (FT-SOAP) is motivated by the success of Ziletti *et al.* FT-SOAP is still under development and has been tested mainly for elemental solids. Also the above-mentioned diffraction fingerprint has been extended while its main limitations remain. In summary, none of these descriptors turned out to provide a significant improvements in comparison with SOAP. As the referee correctly pointed out, there are multiple other descriptors one could test and we also think that this is an important topic that should be addressed in future work.

Moreover, we want to point out that ARISE's internal representation can also be used as novel, data-driven descriptors, that are based on SOAP. In this work, we employ them to explain the model (Fig. 2d-g) or to find hidden patterns in experimental data (Fig. 4b).

1.2) We have investigated several minimum distance parameters and the results are shown and discussed in Supplementary Figure 9a. In line with intuition, for larger minimum distances, points appear more spread in the 2D embeddings. Following the referee's suggestion that the topology might get more interpretable and physically meaningful, we investigated the relationship between embedding and real space. Specifically, we traverse the structure along a circular path in real space (Supplementary Figure 9b, c) and mark the corresponding points in the embeddings according to the angle. We think that in this depiction, it is now much more clear that the cluster shapes are indeed meaningful, as they actually correspond to structural transitions within or between the grains. Besides the changes in Supplementary Figure 9, we have substituted Fig. 2 e-g, which was originally computed for a minimum distance parameter of 0.1, with the corresponding results for a value of 0.9.

1.3) We thank the referee for pointing this out and have removed the grid.

Finally, a few typos I noticed:

(4) p.6, second paragraph: order of words is mixed up in "high-angle annular dark field" (HAADF)

(5) p.8, last paragraph: "obtained via SPM is shown in Fig. 4d" should read "Fig. 4C"

We thank the referee for carefully reading the manuscript and finding the typos, which are now corrected in the new version.

Sincerely,

Wesley F. Reinhart

Reviewer #2 (Remarks to the Author):

This manuscript presents a neural network based approach to assign crystal structure to spatial data. The paper is well written and addresses an important issue. My primary concern with this manuscript is that I'm not sure that the authors have adequately shown the added benefit from this work. The following are my specific comments :

We thank the referee for the assessment and appreciate the acknowledgement of execution and importance of our work. We address the referee's concerns below, in particular trying to clarify the added benefit provided by ARISE with additional explanations and new results.

2.1) In summary, none of the available methods can combine the following points into a single classification model (more detailed explanations and how they address the referee's comments are provided in the other replies):

1. **Flexibility**, i.e., a large number of structural classes. In particular, the correct and robust classification of a diverse range of stacking patterns (fcc – ABC, hcp – AB, dhcp – ABAC http://aflowlib.org/prototype-encyclopedia/A_hP4_194_ac.html , and even α -Sm – ABCBCACAB http://aflowlib.org/prototype-encyclopedia/A_hR3_166_ac.html) has been shown for pristine and highly defective single-crystals (Table 1) and is emphasized by new results added to the revised manuscript (in Fig. 2k-m for a polycrystalline structure published by T. Meiners *et al.* Nature **579**, 375–378, 2020 as well as in Supplementary Figure 12 for several examples from the largest currently available grain boundary database <http://crystalium.materialsvirtuallab.org/>). This diverse range of close-packings cannot be correctly identified by any of the available methods.

2. **Robustness** with respect to missing atoms and atomic displacements — introduced both separately and concurrently. Notably, we provide benchmarking with state-of-the-art methods — a study that to the best of our knowledge has not been performed in comparable fashion in any other publication.

3. **Threshold-independent classification**, i.e., no manual tuning of tolerance parameters is required to arrive at a satisfactory classification.

4. **Uncertainty quantification**, in particular the quantitative role of mutual information — a quantity that is rooted in information theory whose correlation with physical intuition (i.e., structural disorder) is demonstrated in several applications.

5. **Ranking of most similar prototypes**, as determined by the classification probabilities. In particular ARISE's ranking is meaningful not only for the top prediction but also for low-probability candidates in the sense that these contain structural patterns that are similar to the input structure.

6. **Internal neural-network representations**. The usefulness of these internal descriptors has been discussed for slab (Fig. 2d-g) and bulk structures (Fig. 4b). To cover also 2D materials, we have added an analysis for HAADF images (Figure 3d, Supplementary Figure 13), where we use the internal representations to quantify the similarity between images from a defect library (M. Ziatdinov *et al.*, Science Advances **5**, eaaw8989, 2019).

In addition, we have added a study of a quasi-crystalline structure in Fig. 4h,i, which is discussed in the reply 2.2, in the last paragraph.

- The examples shown are either examples where the result could be seen by eye or else where the result is not proven. For example, the simulated data in Figure 2 could have more uncertainty and larger amounts of inconsistency in chemistry or with much more missing data. Then there could be a correct label but with perhaps enough variance that it can't be separated out by eye. Or along this line, the analysis in Figure 3 seems to return what can be seen by eye, and actually AtomNet seems to be doing most of the heavy lifting. And then in Figure 4, the uncertainty change with increased annealing temperature seems obvious as the data will change (particularly as it is tomographically derived) and so I am not sure if that is sufficient for validating the result.

2.2) Regarding the referee's comment "*The examples shown are either examples where the result could be seen by eye or else where the result is not proven.*", we want to emphasize that one of the main goals of ARISE is the correct classification of crystal structures in automatic fashion, without any threshold fine-tuning (e.g., as in spglib). This is discussed in the second paragraph of the introduction, where we state "*Given the large amount of data, the classification should be fully automatic and independent of the manual selection of tolerance parameters (which quantify the*

deviation from an ideal reference structure)." Even if it a structural pattern can be detected by a human expert, the automation of this task is still a remarkable result. The goal is to take the human out of the loop, i.e., even if some of the presented example may be correctly classified by a trained expert, the overall goal of this approach and in general machine learning is often the automation of tedious task, saving valuable human resources. Notably, the correct and automatic classification, especially of bulk structures, is a complex task even for trained materials scientist. In particular, the network is comparing a given input structure to 108 different structural classes. While 2D patterns can be more easy to spot, the classification and detection of defects such as grain boundaries or precipitates in 3D structures is highly non-trivial. In particular, we think that the superalloy model system we consider in Fig. 2h-j addresses this comment: It is highly non-trivial to detect the precipitate and at the same time assign the correct lattice symmetry just by eye. While a trained materials scientist may detect the inclusion in the center, distinguishing the random chemical ordering in the matrix from the ordered L1₂ phase, in particular for more than 20% missing atoms and 5% atomic displacements, is a complex task.

To make this more clear, we have inserted the following sentence in the Discussion:

"Given an unknown structure, the network assigns — in an automatic fashion — the most similar prototypes among 108 possible classes (and quantifies the similarity!), which is a very complicated task even for a trained materials scientist, in particular in case of complex and possibly defective 3D structures."

Regarding 2D image classification, we do think that the grain boundary is non-trivial to spot from the images, which do contain a large level of noise. This is also discussed in the first paragraph of the section *"Application to transmission-electron-microscopy experimental images"*, where we state *"These images contain a substantial amount of noise which makes it very challenging to recognize the graphene honeycomb pattern by naked eye."* To stress this, we have added a further system from a completely different resource — a quasi-crystalline high-resolution transmission electron microscopy (HRTEM) image. We have added two sub-figures (4h,i) and a discussion in the main text (highlighted in red). This structure is aperiodic (no translational symmetry) but does possess a clear order. ARISE does detect the main underlying pattern (pentagons). We think that this example demonstrates how ARISE can be used to classify these complex systems in a systematic and automatic way, pointing out interesting paths for future research. We have hinted at this in the main text at the end of the last paragraph in section *"Application to transmission-electron-microscopy experimental images"*. Already in the original manuscript, the Discussion section hinted at this: *"Since ARISE is not limited to predicting the space group, systems where the space group does not characterize the crystal structure can be tackled (as demonstrated for carbon nanotubes). More complex systems such as quasi-crystals, periodic knots, or weavings could also be considered."*

As a side-note, the section corresponding to Fig. 3 has been re-named from *"Application to STEM experimental images"* to *"Application to transmission-electron-microscopy experimental images"* since with the addition of the quasi-crystal structure, not only STEM images are discussed in this section.

2.3) Regarding the referee's comment *"the simulated data in Figure 2 could have more uncertainty and larger amounts of inconsistency in chemistry or with much more missing data."*, we want to point out that for the slab-system in Fig. 2a, Supplementary Figure 4 contains an investigation of a heavily defective version. This is also mentioned in the main text, in section *"Synthetic polycrystals"*, at the end of the first paragraph: *"The results remain virtually unchanged when introducing atomic displacements (up to 1% of the nearest-neighbor distance) while concurrently removing 20% of the atoms (cf. Supplementary Fig. 4)." Moreover, we test the superalloy model structure for both pristine and heavily defective scenarios in Figure 2j. The mentioned uncertainty in chemistry is also tested in the superalloy system (Fig. 2h-j), since we randomly distribute Ni and Al atoms, which ARISE can distinguish from the ordered precipitate.*

2.4) Regarding the referee's comment "*Or along this line, the analysis in Figure 3 seems to return what can be seen by eye*", we addressed this in the first paragraph of point 2.2). Regarding the comment "*... and actually AtomNet seems to be doing most of the heavy lifting.*", we want to point out that AtomNet is only reconstructing the atomic positions but ARISE is doing the analysis of the crystal structure, i.e., AtomNet alone is not sufficient. Also in the original publication (M. Ziatdinov *et al.*, ACS Nano **11**, 12742, 2017) additional tools have to be employed to analyse the reconstructed atomic positions. Here, we use a robust, threshold-independent, and flexible classification model – a study that has not yet been performed in comparable fashion. See also the additional discussion below (2.8 and Fig. 1).

2.5) Regarding the referee's comment "*And then in Figure 4, the uncertainty change with increased annealing temperature seems obvious [...]*", we want to clarify that the study in Fig. 4c analyses the change in annealing time, not temperature. Moreover, the connection between model uncertainty and physical disorder is intuitively expected but not trivial. It is an important and non-trivial result that an information-theory concept such as mutual information applied to a complex deep-learning model correlates with physical intuition. In particular, this correlation is not obvious. Regarding the statement "*[...] and so I am not sure if that is sufficient for validating the result.*", we want to point out that the correlation of mutual information with the level of defectiveness has also been observed at multiple places for bulk systems at grain boundaries (for instance, Fig. 2c) as well as 2D materials, e.g., STEM images, for both global (Fig. 3a,b) and local studies (Fig. 3g). We thus think that the correlation of mutual information with crystal order is validated. To make this more clear, we have added the following sentence in the discussion: "*This demonstrates the correlation of an information-theory concept with physical intuition.*"

I may be incorrect in what I see as the information gain, and if so that should be clarified. If there was an example shown where it is highlighted what is gained that we didn't have before or some acceleration, etc. that would help emphasize the impact of this work.

2.6) As summarized in point 2.1), we have investigated several additional systems that are all confirmed by experiment or *ab initio* calculations.

- In the AET case, it is a little confusing in what the analysis is picking up. It seems the objective of the approach is to define the crystal structures but here as far as I can tell from what is written is that it is capturing orientation and the structure is all the same. This should be clarified as to what is captured here versus the extent of the approach.

2.7) We want to clarify that in our approach, the orientation of the structure is not the focus, but rather the correct classification of the underlying symmetry, specifically the assignment of the most similar prototype that is known to be found in nature. The most similar prototype is rotationally invariant, the exact orientation of the input structure is not relevant. To include this invariance in a physically motivated way, we use the SOAP representation, which is rotationally invariant. The model trained in this rotation-invariant way is applied to the mentioned AET data.

- Is there any issues when looking at reconstructed, ie. aet, data as opposed to microscopy data?

2.8) The initial input of ARISE are atomic coordinates labelled by chemical species symbols. If no chemical information is available the method will treat the input structure as mono-species.

In general, ARISE can be applied to any microscopy data in which there is a meaningful spatial distribution of particles/clusters whose positions can be reconstructed. Moreover, as shown in the manuscript, ARISE is very robust to noise, which allows to utilize also noisy data as input. Here, we consider microscopy data that provides atomic resolution (i.e., the particles correspond to atoms or

rather atomic columns for bulk systems), in both 2D (STEM HAADF, HRTEM images) and 3D/4D (atomic electron tomography).

For 2D image data, the reconstruction of atoms / atomic columns can be performed due to recent advancements: For high-resolution images, the Python library *atomap* (<https://atomap.org/>) provides routines for fitting 2D Gaussian functions and inferring atomic columns. To support this claim, we apply *atomap* for a selected image – see Fig. 1a below. This image contains two grain boundaries, which are successfully detected via ARISE (within the strided-pattern-matching framework). The most important parameter is the minimum allowed distance for two atoms, which is used for fitting the Gaussian functions. A reasonable choice is to set this parameter to 1 angstrom (while this needs to be specified in units of pixels for *atomap*, cf. Fig. 1b). Notably, artifacts due to fine-tuning of this parameter is mitigated thanks to ARISE's robustness. For instance, increasing the minimum allowed distance to 2 angstrom, will result in a large quantity of missing atoms (cf. Fig. 1c), but ARISE is still capable of detecting the regions of interest. Depending on the application, it may be desirable to choose a larger minimum distance parameter, e.g., if the image contains spurious intensity variations that should not be interpreted as atomic columns.

In the manuscript, we investigated images with high levels of noise for which standard methods such as *atomap* do not provide reasonable reconstructions. Deep-learning frameworks such as AtomNet (where modifications of it are being developed at <https://github.com/pycroscopy/atomai>) can be used to obtain reliable reconstructions, in threshold-independent manner.

The importance of ARISE's robustness is also mentioned in the manuscript, in the section "*Application to transmission-electron-microscopy experimental images*", end of first paragraph: "*It is also important to point out that ARISE robustness well beyond physical levels of noise is essential to achieve the correct classification despite the presence of substantial amount of noise from both experiment and atomic position reconstruction.*"

Figure 1:

a Aberration-free transmission-electron-microscopy image of gold atoms, taken from <https://www.nanotech-now.com/products/nanonewsnow/issues/034/034.htm>. Atomap (<https://atomap.org/>) is employed to reconstruct the atomic columns (left part of **b**, **c**). The extracted 2D coordinates can be analyzed using ARISE in the strided-pattern-matching (SPM) framework, revealing, in automatic fashion, the region containing the grain boundaries (right part of **b**, **c**). In particular, the exact choice for the minimum allowed separation in atomap, which can lead to various atoms being missed, is not relevant for the detection of the grain boundaries due to the robustness of ARISE.

- It is a different technique, but is mentioned in the introduction: in the standard APT analysis software (such as through IVAS), the bond distributions are a standard input as well as different clusters and some approximation of structure. Mention of what is added here compared to that would be helpful.

2.9) For a given structure, ARISE does not make any assumptions on clusters or any structural content. It is trained on pristine prototypes that do not contain any information on polycrystalline structures such as investigated in APT experiments. In this sense, ARISE is much more unbiased than IVAS. The application of ARISE to APT experiments is an interesting idea on which we are currently working. However, a discussion and comparison to existing APT data analysis software is beyond the scope of this paper.

- The term 'deep learning' is used a lot now, and I think justification of how this is deep learning and not machine learning (especially since the neural network used isn't convolutional neural network) would be good.

2.10) We respectfully disagree with the referee's assessment that it is not clear in which sense our method qualifies as deep learning, in particular the statement "*especially since the neural network used isn't convolutional*". The type of model, a fully connected neural network (multilayer perceptron involving affine and non-linear transformations), fully qualifies as deep neural network.

Indeed, the use of convolutional layers (like in a convolutional neural network) is completely irrelevant to the fact that a model is considered deep learning or not. We refer to <https://www.deeplearningbook.org/>, especially chapter 6, for more details.

Thus, it is fully justified to refer to our work as deep learning, both in architecture type (involving affine and non-linear transformations) and model complexity (~370k parameters, three hidden layers with up to 512 neurons, cf. Table 2 for the detailed architecture).

We also want to remark that multilayer perceptrons are regularly employed in industry and science: for instance, multilayer perceptrons are important building blocks for representing the many-body wave function, cf. Pfau *et al.* Physical Review Research 2, 033429 (2020).

Notably, ARISE is a Bayesian neural network, i.e., an extension of the standard fully connected architecture.

- On page 4, start of the first paragraph, it is set that 80% of data is used for training and the rest for test. Is this randomly selected and evenly distributed? Also, in experimental data we often have large regions where the data is not evenly represented. How does the result change if the training is biased to specific regions?

2.11) The splitting is performed in random and stratified fashion using scikit-learn (specifically https://scikit-learn.org/stable/modules/generated/sklearn.model_selection.train_test_split.html).

The stratification ensures that after splitting, each class is represented in both splits by the same fractional amount as in the original, total dataset. This is a standard procedure, especially in problems with a large number of classes like in our case (108 classes). This is discussed in the Methods, subsection “*Neural network architecture and training procedure*”, third paragraph: “*The initial training set is split (80/20% training/validation split of pristine structures, performed using scikit-learn and a random state of 42) ...*”, where we have added “*in stratified fashion*”.

2.12) Regarding the question “*How does the result change if the training is biased to specific regions?*” we first want to clarify that no structural regions from experiment are employed but only pristine structures that are known to be found in nature. Performing an unstratified split and including only few examples for certain classes would lead to poor generalization ability for these systems, since the model has only seen few examples during training. The fact that ARISE – although only trained on pristine synthetic data – generalizes to a large variety of systems from computations and experiments is one of the strongest results of the proposed approach. This is also mentioned in the Discussion: “*ARISE is trained on ideal synthetic systems only and correctly identifies crystal structures in STEM and AET experiments, hence demonstrating strong generalization capabilities.*”

Reviewer #3 (Remarks to the Author):

The manuscript presents a method that employs neural networks to classify atomic structures. Trained on a synthetic database where the entries were labelled, the authors present a number of benchmark tests where the model is capable of accurately predicting the appropriate label. Applications on experimental results are also shown.

The topic is interesting and the work is solid, but overall it is let down by the poorly written text, especially in the first half of the paper. I list a couple of remarks and examples of imprecise prose.

- The first paragraph of the introduction is somewhat difficult to comprehend.

- “NNs are non-linear classifiers”: not true, they can be used as classifiers as well as regressors etc.

- "distortions beyond the physical level": I am not even sure what a distortion at the physical level may be
- "Standard NNs are unable to provide reliable model classification uncertainty. There is a widespread use of the probability provided by the last layer as uncertainty estimate. Typically, the so-called softmax function is employed to normalize the sum of output values such that they can be interpreted as classification probabilities.": how do these sentences belong together?

We thank the referee for the comments. In particular, we appreciate that the referee acknowledges the relevance and quality of our work.

Regarding the mentioned imprecise formulations, we thank the referee for carefully reading the manuscript and have replaced the mentioned sentences and paragraphs in the following way:

3.1) *"Identifying the crystal structure of a given material is important for understanding and predicting its physical properties. For instance, the hardness of industrial steel is strongly influenced by the atomic composition at grain boundaries, which has been studied in numerous theoretical and experimental investigations. Beyond bulk materials, two- (2D) and one-dimensional (1D) systems have far-reaching technological applications, such as solar energy storage, DNA sequencing, cancer therapy, or even space exploration. To characterize the crystal structure of a given material, one may assign a symmetry label, e.g., the space group. More generally, one may want to find the most similar structure within a list of given known systems. These structural classes are identified by stoichiometry, space group, number of atoms in the unit cell, and location of the atoms in the unit cell (the Wyckoff positions)."*

3.2) *"Neural networks are non-linear machine-learning models."*

3.3) *"[...] e.g., distortions beyond a level that can be explained by a physical effect or, [...]"*

3.4) *"Standard NNs are unable to provide reliable model uncertainty. In a classification setting, there is widespread use of the probability provided by the last layer as uncertainty estimate. These probabilities are typically obtained by normalizing the sum of output values using the so-called softmax activation function"*.

We also point out that the use of the last layer of a neural network as probability estimate is standard procedure within neural network practitioners since many years. For a short – yet comprehensive review on deep learning – one can refer for example to LeCun *et al.*, Nature **521**, 436 (2015), or any introductory deep-learning book.

Regarding the method, one may ask if using a neural network is really necessary. There are numerous Bayesian classifier methods which do not use NNs, it would be interesting to know whether these were considered but rejected or NNs were chosen mainly for the hype factor. Otherwise the methodology is based on some well-established tools, combined with materials databases.

3.5) We think that the application of deep learning is well justified given the high-dimensionality of the input (316 SOAP components) combined with the complexity of the classification problem (108 structural classes covering 1D, 2D, and bulk materials) as well as the fact that deep neural networks are known for their superior performance especially for classification tasks (see, for instance Y. LeCun *et al.*, Nature **521**, 436, 2015). Certainly, it is important to keep in mind other machine learning methods such as the mentioned Bayesian classifiers. To address this, we have tested the Naive Bayes (NB) method (as implemented in the Python library scikit-learn) where two different flavors of NB are tested that differ by the way the likelihood $p(x_i|y)$ of the features x_i given the label y is defined: Gaussian NB, assuming a Gaussian likelihood, and Bernoulli NB, assuming a multivariate Bernoulli distribution. We train both NB methods using exactly the same training and test set that was used for ARISE. Bernoulli NB shows the best performance with $\sim 76\%$ accuracy on pristine structures and gradually decreasing performance for defective structures. We have added two additional lines in Table 1. Compared to hand-crafted methods such as spglib, CNA, a-CNA, BAA, and PTM, Bernoulli and Gaussian NB are a clear improvement in terms of the number of

classes. However, both Bernoulli and Gaussian NB are significantly surpassed by ARISE which classifies pristine and defective structures with 100% classification accuracy – a number that starts to gradually decrease only at high levels of noise (beyond 10% missing of atoms or 1% atomic displacements).

Considering the applications, I found those presented in Fig. 2 indeed synthetic, as these were not even applied on results of atomic simulations but constructed for demonstrations. I would have been nice to see more realistic scenarios covered here. In the interpretation and classification of experimental data, these are essentially 2D problems and I would be surprised if the standard image recognition machinery hasn't been applied to them. It would be interesting to see how ARISE extracts more information and provides new understanding.

3.6) To address the referee's comment on more realistic scenarios, we have investigated two examples for simulated data:

- Fig. 2k-m (plus discussion in the main text, highlighted in red): A system from an evolutionary structure search that has been validated by STEM experiments (T. Meiners *et al.* Nature **579**, 375, 2020). Two fcc grains are separated by a grain boundary with a peculiar pattern (termed “pearl” by the authors). ARISE assigns the dhcp prototype at the grain boundary regions which is validated by visual inspection: The assignments are meaningful as the local regions at the grain boundary partially consist of fcc while containing distortions and changed stacking. Since dhcp can be understood as a 50/50 mixture of fcc and the lower symmetry phase hcp (http://afflowlib.org/prototype-encyclopedia/A_hp4_194_ac.html), this assignment is meaningful.
- Supplementary Figure 12: To confirm the previous point, an extraction of systems from the currently largest available grain boundary database is analyzed. The representative systems from the database cover multiple symmetries (fcc, bcc, hcp, dhcp) and grain boundary types (twist, tilt). ARISE can classify correctly multiple symmetries, including complex stacking patterns such as dhcp (ABAC) and not only fcc or hcp — one of the major distinct characteristics that distinguish ARISE from other available methods. Moreover, ARISE can even correctly identify more complex packings such as the α -Sm type (ABCBCACAB, http://afflowlib.org/prototype-encyclopedia/A_hr3_166_ac.html).

3.7) Regarding the referees comment “[...] *In the interpretation and classification of experimental data, these are essentially 2D problems and I would be surprised if the standard image recognition machinery hasn't been applied to them [...]*” : In image recognition, deep neural networks have emerged as the state-of-the-art approach, outperforming standard image recognition methods. This has been demonstrated by the record-breaking performance on the ImageNet database that outperformed all other standard approaches (A. Krizhevsky *et al.* Advances in Neural Information Processing Systems **25**, 1097, 2012). The superiority of deep learning in this context has been confirmed since then in multiple other settings (see, for instance, the review Y. LeCun *et al.* Nature **521**, 436, 2015). Motivated by this success, deep neural networks have been adapted to materials science problems, including electron microscopy data. A very recent review (J. M. Ede, Machine Learning: Science and Technology **2**, 011004, 2021) provides a comprehensive overview. Especially sections “1.3 Labelling” and “1.4 Semantic segmentation” mention several methods that analyze atomic-resolution images and classify its crystal structure, the most notable being

- J. Aguiar *et al.* Science Advances **5**, eaaw1949 (2019): Space group prediction based on electron diffraction fingerprint or the fast Fourier transform (FFT) of a high-resolution image.
- R. K. Vasudevan *et al.* npj Computational Materials **4**, 1 (2018): Bravais lattice classification based on FFT of high-resolution image.

Both of these methods perform a global assignment of a symmetry label to a given image. For the local analysis of images, approaches have been developed to identify defects (A. Maksov *et al.* *npj Computational Materials* **5**, 1, 2019) or atomic positions (M. Ziatdinov *et al.* *ACS Nano* **11**, 12742, 2017). Notably, these approaches require further analysis tools for the classification of the crystal structure, e.g. via unsupervised learning (A. Maksov *et al.* *npj Computational Materials* **5**, 1, 2019) or graph-based methods (M. Ziatdinov *et al.* *ACS Nano* **11**, 12742, 2017).

ARISE can be used for both local and global analysis. This is discussed, for instance, in section “*Application to transmission-electron-microscopy experimental images*”, in the first paragraph where we state “[...] we show how global and local analysis can be combined to analyze crystal structures.” and in the second to last paragraph: “*While so far we have analyzed HAADF images on a global scale, a local analysis via SPM allows to zoom into a given structure and locate sub-structural features.*”

Moreover, none of the above mentioned approaches offers a comparable flexibility in terms of the number of classes that can be correctly identified – for instance, we extract an exhaustive selection of 2D materials from the currently largest available repository <https://cmr.fysik.dtu.dk/c2db/c2db.html> which also includes systems where no meaningful space group or Bravais lattice can be identified.

In summary, none of the available methods can combine the following points into a single classification model:

1. **Flexibility**, i.e., a large number of structural classes. In particular, the correct and robust classification of a diverse range of stackings (fcc – ABC, hcp – AB, dhcp – ABAC http://aflowlib.org/prototype-encyclopedia/A_hp4_194_ac.html, and even α -Sm – ABCBCACAB http://aflowlib.org/prototype-encyclopedia/A_hR3_166_ac.html) has been shown for pristine and highly defective single-crystals (Table 1) and is emphasized by new results added to the revised manuscript (in Fig. 2k-m for a polycrystalline structure published by T. Meiners *et al.* *Nature* **579**, 375–378, 2020 as well as in Supplementary Figure 12 for several examples from the largest currently available grain boundary database <http://crystalium.materialsvirtuallab.org/>). This diverse range of close-packings cannot be correctly identified by any of the available methods.
2. **Robustness** with respect to missing atoms and atomic displacements — introduced both separately and concurrently. Notably, we provide benchmarking with state-of-the-art methods — a study that has not been performed in comparable fashion in any other publication.
3. **Threshold-independent classification**, i.e., no manual tuning of tolerance parameters is required to arrive at a satisfactory classification.
4. **Uncertainty quantification**, in particular the quantitative role of mutual information — a quantity that is rooted in information theory whose correlation with physical intuition (i.e., structural disorder) is demonstrated in several applications.
5. **Ranking of most similar prototypes**, as determined by the classification probabilities. In particular ARISE’s ranking is meaningful not only for the top prediction but also for low-probability candidates in the sense that these contain structural patterns that are similar to the input structure.
6. **Internal neural-network representations**. The usefulness of these internal descriptors has been discussed for slab (Fig. 2d-g) and bulk structures (Fig. 4b). To cover also 2D materials, we have added an analysis for HAADF images (Figure 3d, Supplementary Figure 13), where we use the internal representations to quantify the similarity between images from a defect library (M. Ziatdinov *et al.*, *Science Advances* **5**, eaaw8989, 2019). We discuss this in more detail in the next point (3.8).

3.8) The referee stated that *“it would be interesting to see how ARISE extracts more information and provides new understanding.”* In point 3.7), we have listed several aspects of unique information that ARISE provides, supported by studies on simulated grain-boundary structures.

One aspect of ARISE is its internal, data-driven representations that are learned during training. In the manuscript this has been used in the analysis of bulk structures (Fig. 2d-g and 4b). To demonstrate the usefulness of these representations also for 2D data, we conducted the following study (adding sub-figure 3d, a discussion in the main text, and Supplementary Figure 13):

We demonstrate how one can use these representations to quantify the similarity between STEM images. Before, in Fig. 3a-c, we used ARISE to quantify the similarity of a single image to the training structures. Thus, this analysis is clearly addressing a new topic. Specifically, given two images, we extract the corresponding representations from the hidden layers of the neural network. For instance, for the central layer, this would give two vectors of dimensionality 512. In Fig. 3d, the similarity between the structures shown in Fig. 3a,e and several Si-doped graphene systems is investigated. As discussed in more detail in the main text, a block-matrix structure arises in which the binary and mono-species structures are separated. Supplementary Figure 13 performs a similar analysis but for the full database of Si-graphene structures. The block matrix structure is preserved, which supports our previous analysis on a sub-selection of structures. This analysis can be performed for any other structure, in any dimension (i.e., for 1D, 2D or bulk systems, for pristine or defective structures).

This study confirms that ARISE learns meaningful similarity concepts, which go beyond the (already complex) classification task, providing new information obtained thanks to ARISE’s data-driven approach.

In addition, we have added a study of a quasi-crystalline structure in Fig. 4h,i alongside discussion in the main text. This structure is from a completely different resource — a quasi-crystalline high resolution transmission electron microscopy (HRTEM) image. Notably, the structure is aperiodic (no translational symmetry) but does possess a clear order. ARISE does detect the main underlying pattern (pentagons). We think that this example demonstrates how ARISE can be used to classify these complex systems in a systematic and automatic way, pointing out interesting paths for future research. We have hinted at this in the main text at the end of the last paragraph in section *“Application to transmission-electron-microscopy experimental images”*. Already in the original manuscript, the Discussion section hinted at this: *“Since ARISE is not limited to predicting the space group, systems where the space group does not characterize the crystal structure can be tackled (as demonstrated for carbon nanotubes). More complex systems such as quasi-crystals, periodic knots, or weavings could also be considered.”*

As a side-note, the section corresponding to Fig. 3 has been re-named from *“Application to STEM experimental images”* to *“Application to transmission-electron-microscopy experimental images”* since with the addition of the quasi-crystal structure, not only STEM images are discussed in this section.

Overall, the paper in its current form cannot be accepted for publication in Nature Communications. The text needs attention and the results strengthening. Following a major revision, it should be revisited by the reviewers to see if ARISE indeed delivers.

We believe that we have sufficiently addressed the referee’s comments and are looking forward to further exchange.

REVIEWERS' COMMENTS

Reviewer #1 (Remarks to the Author):

The authors have addressed my comments from the first review. I recommend the article be published.

Reviewer #2 (Remarks to the Author):

The changes made addresses my concerns adequately. I would recommend for publication.

Reviewer #3 (Remarks to the Author):

The authors have satisfactorily addressed the Referees' concerns and have significantly improved the text. The manuscript is publishable.